# The impact of mating and sugar feeding on blood-feeding physiology and behavior in the arbovirus vector mosquito *Aedes aegypti*

**Garrett P. League**[1], **Ethan C. Degner**[1¤], **Sylvie A. Pitcher**[1], **Yassi Hafezi**[2], **Erica Tennant**[1], **Priscilla C. Cruz**[1], **Raksha S. Krishnan**[1], **Stefano S. Garcia Castillo**[3], **Catalina Alfonso-Parra**[4,5], **Frank W. Avila**[5], **Mariana F. Wolfner**[2]*, **Laura C. Harrington**[1]*

**1** Department of Entomology, Cornell University, Ithaca, New York, United States of America, **2** Department of Molecular Biology and Genetics, Cornell University, Ithaca, New York, United States of America, **3** Laboratorio de Malaria: Parásitos y Vectores, Laboratorios de Investigación y Desarrollo, Facultad de Ciencias y Filosofía, Universidad Peruana Cayetano Heredia, Lima, Peru, **4** Instituto Colombiano de Medicina Tropical, Universidad CES, Sabaneta, Antioquia, Colombia, **5** Max Planck Tandem Group in Mosquito Reproductive Biology, Universidad de Antioquia, Medellín, Antioquia, Colombia

¤ Current address: Biology Department, Wisconsin Lutheran College, Milwaukee, Wisconsin, United States of America

* mariana.wolfner@cornell.edu (MFW); lch27@cornell.edu (LCH)

**Data Availability Statement:** All relevant data are within the manuscript and its Supporting Information files.

## Abstract

### Background

*Aedes aegypti* mosquitoes are globally distributed vectors of viruses that impact the health of hundreds of millions of people annually. Mating and blood feeding represent fundamental aspects of mosquito life history that carry important implications for vectorial capacity and for control strategies. Females transmit pathogens to vertebrate hosts and obtain essential nutrients for eggs during blood feeding. Further, because host-seeking *Ae. aegypti* females mate with males swarming near hosts, biological crosstalk between these behaviors could be important. Although mating influences nutritional intake in other insects, prior studies examining mating effects on mosquito blood feeding have yielded conflicting results.

### Methodology/Principal findings

To resolve these discrepancies, we examined blood-feeding physiology and behavior in virgin and mated females and in virgins injected with male accessory gland extracts (MAG), which induce post-mating changes in female behavior. We controlled adult nutritional status prior to blood feeding by using water- and sugar-fed controls. Our data show that neither mating nor injection with MAG affect *Ae. aegypti* blood intake, digestion, or feeding avidity for an initial blood meal. However, sugar feeding, a common supplement in laboratory settings but relatively rare in nature, significantly affected all aspects of feeding and may have contributed to conflicting results among previous studies. Further, mating, MAG injection, and sugar intake induced declines in subsequent feedings after an initial blood meal, correlating with egg production and laying. Taking our evaluation to the field, virgin and mated mosquitoes collected in Colombia were equally likely to contain blood at the time of collection.

**Funding:** This study was supported by a National Institute of Allergy and Infectious Diseases (www.niaid.nih.gov) grant R01AI095491 to LCH and MFW, as well as a Colciencias (minciencias.gov.co), Universidad de Antioquia and Max Planck Society cooperation grant 566-1-2014 to FWA. ECD received support from a Cornell University Graduate School Sage Fellowship (gradschool.cornell.edu/financial-support/fellowships/new-student-fellowships) and a Cornell University Einaudi Center International Research Travel Grant (einaudi.cornell.edu/funding/travel-grants). SSGC received support from a Fogarty International Center (www.fic.nih.gov) Translational Research Development for Endemic Infectious Diseases of Amazonia grant 2D43TW007120-11A1 to Dr. Joseph M. Vinetz and Dr. Dionicia Gamboa Vilela. The funders had no role in study design, data collection and analysis, decision to publish, or preparation of the manuscript.

**Competing interests:** The authors have declared that no competing interests exist.

## Conclusions/Significance

Mating, MAG, and sugar feeding impact a mosquito's estimated ability to transmit pathogens through both direct and indirect effects on multiple aspects of mosquito biology. Our results highlight the need to consider natural mosquito ecology, including diet, when assessing their physiology and behavior in the laboratory.

### Author summary

Controlling mosquitoes and the disease agents they transmit during blood feeding remains a global public heath priority. Some vector control tools in development target the mosquito reproductive system as a means of control. The transfer of semen to a female during mating profoundly impacts her biology by inhibiting re-mating and stimulating egg production and laying. Previous studies in *Aedes aegypti* that examined potential effects of mating on blood feeding yielded contradictory results. We examined the potential of mating and seminal fluids to modulate female blood feeding and controlled for numerous experimental factors, such as ingestion of sugar prior to blood feeding, that may have led to conflicting results in prior studies. In laboratory studies with field-derived Thai mosquitoes, we show that mating and seminal fluids do not impact blood feeding during a mosquito's first meal, but prior sugar feeding does. Furthermore, we show that mating, seminal fluids, and sugar feeding can impact blood feeding across multiple consecutive meals. Furthermore, field studies in Colombia show that virgin and mated females were equally likely to contain blood. Our research clarifies the impact of mating on blood feeding and suggests ways to improve our understanding of these behaviors in nature.

## Introduction

Mosquito-borne diseases continue to pose a major public health threat across the globe, particularly in resource-poor tropical regions where transmission levels are often highest. *Aedes aegypti* mosquitoes are a major vector of arthropod-borne viruses (arboviruses) that cause diseases such as dengue, yellow fever, Zika, and chikungunya [1–3]. As *Aedes* vector populations, many of which are insecticide resistant [4], continue to expand their ranges at historic rates [5], so too does the spread of the pathogens they harbor [6]. As an anautogenous species [7], *Ae. aegypti* females transmit pathogens between human hosts and acquire essential nutrition for egg production and energy during blood feeding. Hence, understanding the biological factors that modulate blood feeding in *Ae. aegypti* could considerably enhance our ability to understand disease transmission dynamics.

Mating can have profound downstream physiological and behavioral effects in mosquitoes. In *Ae. aegypti*, the transfer of male accessory gland fluid (i.e., seminal fluid) proteins to females during mating renders females resistant to subsequent male mating attempts [8–15], increases egg development and oviposition [16–26], inhibits courtship acoustic harmonization [27], decreases flight activity [28–33], reduces host seeking behavior [19,34–39], increases survival [40], and induces immune responses [41,42]. While the precise timing of most of these effects are still unknown, refractory behavior appears quickly, within two hours of mating [8]. In other organisms, such as the fruit fly *Drosophila melanogaster*, mating also induces a wide range of well-characterized female post-mating responses, including changes in mating

receptivity, ovulation, longevity, immunity, and feeding habits [43]. With respect to feeding, a male seminal fluid protein causes mated *Drosophila* females to ingest substantially larger meals than virgins, a behavior that provides increased nutritional stores for egg production [44]. In nature, *Ae. aegypti* mate in single pairs or in small swarms that form in response to chemosensory cues emitted by human hosts [45,46]. In these swarms, males fly in characteristic figure-eight "patrolling" patterns near hosts [46,47], increasing their odds of finding a host-seeking female [39,48]. As females that approach hosts to blood feed are often intercepted and inseminated by swarming males either before or after obtaining a blood meal [49], researchers have long suspected potential biological crosstalk between mating and blood feeding. However, our understanding of the relationship between mating and blood feeding in mosquitoes suffers from several longstanding and unresolved discrepancies.

Previous studies in mosquitoes that examined potential effects of mating on various aspects of blood-feeding physiology and behavior yielded conflicting results (reviewed in [47,50–52]), likely owing to variation in a number of key experimental parameters (Table 1). Seaton and Lumsden found no difference between virgin and mated female blood feeding avidity, which is defined as a mosquito's propensity to take a blood meal [53]. Early studies by Lavoipierre in *Ae. aegypti* [37] documented similar blood feeding avidity in virgin and mated females in an initial blood meal, but diminished blood feeding in mated females at the onset of egg laying, which occurred predominantly in mated females. Subsequent work by Judson [19] confirmed that this feeding pattern was dependent upon male accessory gland extract (MAG). With respect to blood meal digestion, Edman [54] found that mated females digested blood more rapidly than virgins, a finding which was later reproduced in MAG-treated females in experiments conducted by Downe [55]. However, these studies did not include water-fed female controls and did not restrict access to sugar to the initial post-eclosion period, likely leading to higher levels of sugar feeding than are experienced by *Ae. aegypti* in nature.

Adlakha and Pillai [56] were the first to observe an effect of mating on the size (weight) of an initial blood meal, finding that both *Ae. aegypti* and *Culex pipiens* females ingested more blood after mating with males with intact male accessory glands compared to virgins and females mated with males whose accessory glands were surgically removed. However, a follow-up study by Klowden [57] contradicted these results by accounting for fluid loss via diuresis, the rapid fluid excretion that occurs during and after blood feeding [61,62]. By weighing females immediately post-feeding and using the diuresis-independent hemiglobincyanide (HiCN) technique, which employs Drabkin's reagent to convert hemoglobin to the colorimetric product HiCN [63], he showed that virgin and mated females ingested identical amounts of blood during an initial feeding. However, a later study by Houseman and Downe [58] introduced new uncertainty to this question, finding that mated females ingested more blood than virgins at six days post-eclosion, but not three days post-eclosion, when measuring blood meal size using the Bramhall technique (a colorimetric protein assay), but not the HiCN technique. These authors also found that mating shortly after eclosion increased blood meal consumption approximately 10 days later (an unlikely time interval between mating and blood feeding in nature [64–68]) using both assays in an initial experiment, but using only the Bramhall technique in a replicate experiment. Building on earlier observations of accelerated blood meal digestion in mated and MAG-injected females by Downe [55], they also found evidence that mating increases blood meal digestion by accelerating the rate of protein loss through increased proteinase activity. As with previous studies, differences in access to sugar, diuresis control, mosquito age, blood meal measurement methods, experimental replication, and mosquito strain all likely contributed to these discrepancies.

Using a recently colonized Thai strain of *Ae. aegypti*, Villarreal et al. [40] documented trends suggestive of increased blood meal size during an initial feeding and increased feeding avidity

**Table 1. Summary of previous studies on the effect of mating on blood-feeding avidity, digestion, and intake in mosquitoes.**

| Publication | Mosquito | Strain | Adult nutrition | Timing of mating PE | BM source | Duration of feeding | Timing of feeding PE | Mating to BM interval | Assay | Major finding(s) |
|---|---|---|---|---|---|---|---|---|---|---|
| Seaton and Lumsden, 1941 [53] | *Ae. aegypti* | Liverpool | No sugar or water | <1, 1 and 2 d PE | Human forearm | 10 min | 3–4 d PE | 1–4 d | Feeding avidity | Virgin and mated female feeding avidities are similar |
| Lavoipierre, 1958a [37] | *Ae. aegypti* | Liverpool | Likely sugar (based on [38]) | ≤2 d PE [37] | Human | 5 min | 2 d PE [37] | 1–2 d [37] | Feeding avidity and oviposition | Virgin and mated female feeding avidity is similar in an initial BM but mated female feeding avidity decreases with onset of oviposition cycle |
| Edman, 1970 [54] | *Ae. aegypti* | New Orleans | 10% sucrose except 24 h before and after bm | Held with males PE | Human forearm | 30 min | 6 d PE | 4–5 d | Precipitin protein detection test | Mated females digest blood faster than virgins |
| Downe, 1975 [55] | *Ae. aegypti* | NIH | 10% sucrose | Held with males for 96 h PE | Human forearm (for precipitin test) or rabbit (or immunodiffusion test) | To engorgement | 5 d PE | 1–5 d | Precipitin and immunodiffusion tests | Mated and MAG-treated females digest blood faster than virgins |
| Adlakha and Pillai, 1976 [56] | *Ae. aegypti* and *Culex pipiens* | Delhi | 1% glucose for first 48 h PE | 2 d PE | Guinea pig (*Ae. aegypti*) or pigeon (*Cx. pipiens*) | 1 h (*Ae. aegypti*) or overnight (*Cx. pipiens*) | 4 d PE | 2 d | Weight | Increased BM size in mated females compared to virgins |
| Klowden, 1979 [57] | *Ae. aegypti* | Unknown | 1% glucose for first 48 h PE then starved for next 48 h | Reared with males | Rat | To engorgement | 4 d PE | 1–4 d | Weight, HiCN | No change in BM size PM |
| Houseman and Downe, 1986 [58] | *Ae. aegypti* | Queen's | 20% sucrose (based on [59]) | Reared with males | Guinea pig | 20 min | 3, 6, and 10 d PE | ≤3 d, ≤6 d, and ≤10 d | HiCN, trypsin activity | No change in BM size in females 3 d PE; mixed results in females at 6 & 10 d PE. Delayed onset of BM digestion in virgins. |
| Villarreal et al., 2018 [40] | *Ae. aegypti* | Thai | 10% sucrose, removed 24 h before BM | 3–5 d PE mated/ injected w/ MAG | Human forearm | 20 min | 7–9 d PE | 3–4 d | Weight | Increased BM size/feeding propensity PM in some, but not all trials |
| Dahalan et al., 2019 [60] | *An. coluzzii* | Ngousso | 10% fructose | Held with males overnight PE | Human blood (artificial membrane) | 10 min | 1 d PE | ≤1 d | Weight, HiCN | Virgin and mated females ingest same amount of blood and digest at similar rates |

*(Continued)*

**Table 1.** (Continued)

| Publication | Mosquito | Strain | Adult nutrition | Timing of mating PE | BM source | Duration of feeding | Timing of feeding PE | Mating to BM interval | Assay | Major finding (s) |
|---|---|---|---|---|---|---|---|---|---|---|
| Current study: | *Ae. aegypti* | Thai (lab), Colombian (field) | 10% sucrose from eclosion to mating/ injection or no sugar | 1–3 d PE mated/ injected w/ MAG (lab) | Human forearm or whole human host (lab) | 2–5 min (lab) | 2–6 d PE (lab) | 1–3 d (lab) | Weight, hemoglobin assay, behavioral assays (+field collections) | Mated females ingest, digest, and feed upon blood at similar rates. |

Publications are presented in chronological order. Abbreviations: PE, post-eclosion; BM, blood meal; MAG, male accessory gland extract; HiCN, hemiglobincyanide method; PM, post-mating.

across multiple feedings in mated females and MAG-injected virgin females compared to virgins. However, these trends were inconsistent across trials, feedings took place at room temperature, and the experiments did not include water-fed controls (i.e., water alone, with no sugar added), with females provided continuous access to sugar except for the day before each blood feeding. In *Anopheles* mosquitoes, Dahalan et al. [60] found identical amounts of blood in virgin and mated mosquitoes both immediately and 24 h post-feeding using both weighing and HiCN methods, suggesting that mating did not affect blood meal size or digestion rates of an initial meal. Beyond experiments like these, which were conducted in the laboratory, few blood feeding parameters have been thoroughly examined with respect to mating status in field mosquitoes.

To understand the effects of mating on female blood-feeding biology in a comprehensive manner, we tested whether mating and MAG regulate blood meal size, digestion, and feeding avidity in *Ae. aegypti* using gravimetric, chemical, and behavioral assays. To resolve previous inconsistencies, we tested the hypothesis that variation of important, but often uncontrolled experimental parameters, such as continuous sugar provision, influenced the conflicting results obtained from prior studies. Our experiments controlled for these potentially confounding factors by using field-relevant *Ae. aegypti* (Thai and Colombian), employing only human blood meals, standardizing mosquito sizes and ages, controlling for diuresis, carefully monitoring ambient temperature and humidity, and, importantly, accounting for sugar feeding prior to blood feeding, which is rare in arbovirus endemic regions [69–75] but can alter the outcome of laboratory blood-feeding experiments [75,76]. Since *Ae. aegypti* naturally ingests multiple blood meals within a single gonotrophic cycle [77,78], we also conducted experiments comparing feeding avidities across multiple blood meals. Although this approach is not typically done, it is important when investigating mosquito feeding behavior. Finally, we compared our laboratory-based findings to mating and blood meal status data from field-collected mosquitoes. Overall, this study demonstrates that both mating (across multiple consecutive blood meals, but not in an initial meal) and sugar feeding (across multiple blood meals as well as in an initial meal) reduce a female's tendency to blood feed, with important implications for mosquito vectorial capacity and disease transmission.

## Methods

### Mosquito rearing and maintenance

A Thai strain of *Ae. aegypti* mosquitoes originating from field-collected populations in Bangkok, Thailand (15˚72'N, 101˚75'E) was used for all laboratory experiments. This colony has

been supplemented annually with $F_1$ eggs since 2009 to maintain natural levels of genetic heterogeneity. Individual trials for each set of experiments represented true biological replicates that originated from a separate generation of eggs produced by the Thai colony. Adults arising from colony eggs were held in separate 8 L plastic bucket cages for each experimental trial to avoid cage effects. Mosquito maintenance and all blood feedings for experiments were conducted in an environmental chamber at 26.83 ± 0.27˚C and 85.09 ± 4.60% RH as previously described [40]. Mosquito rearing was performed to obtain medium-sized adult females [13,27]. To verify uniform female body size across experiments (S1 Fig), wings were collected from a subset of the mosquitoes after each experiment (wing lengths: 2.86 ± 0.09 mm; n = 629) and measured as previously described [79] using cellSens standard software v1.17 (Olympus Corp., Shinjuku City, Tokyo, Japan).

### Sugar feeding treatments

Sugar water treatment group ("Sugar") adults were provided with 10% (1 g/10 mL) sucrose prepared using distilled water and granulated sugar (Best Yet, Keene, NH, USA or similar product) for 24 h preceding and following experimental mating treatment for blood meal size and digestion experiments (48 h total), and for 24 h prior to treatment alone for all feeding avidity experiments (24 h total; for details, see S2 Fig and "Blood-feeding avidity measurements" subsection below). After the initial feeding period, sugar water was replaced with distilled water alone for the remainder of the experiment to mimic natural feeding habits [69,76]. Water alone treatment group ("Water") adults were fed exclusively on distilled water. For all experiments, females fed on sugar or water sources *ad libitum*.

### Female injection and mating treatments

Injection and mating treatment for males and females occurred at 2–3 days post-eclosion (PE) for blood meal size and digestion experiments and 1–2 days PE for blood-feeding avidity experiments (S2 Fig). Males were anesthetized briefly on ice and their accessory glands, with seminal vesicles attached, were dissected, processed, and injected into virgin females ("MAG" treatment group) as a saline-based homogenate (MAG) as previously described [27]. Verification of the equivalency of MAG concentrations (1.06 ± 0.15 ug/ul ± SD; n = 11) between experimental trials was performed as previously described [27]. To control for effects on blood feeding due to injection itself, a cohort of virgin females was injected with PBS alone ("Saline"). Injected females, as well as non-injected virgin ("Virgin") and mated ("Mated") females, were placed in 2 L cardboard recovery buckets and transferred to the environmental chamber described above. For mated groups, we added males to females buckets at a 1:1 male to female ratio for two days in blood meal size and digestion experiments and at a 2:1 male to female ratio for one day in feeding avidity experiments (S2 Fig). Sex ratios and holding times were based on previous work in our lab [40] and pilot experiments confirmed that they yielded 100% female-mating rates prior to blood feeding. For these reasons, as well as other experimental considerations (e.g., the need to preserve female abdomens for hemoglobin measurements; see below), we did not perform spermathecal dissections to verify mating status after blood meal size and blood meal digestion experiments, or in some of our feeding avidity experiments (initial and multiple blood meal experiments on human host forearm). After these mating periods, males were removed from mated group buckets prior to blood feeding to allow for female recovery and feeding in the absence of male harassment. Mosquitoes were held for a total of one to three days prior to blood feeding and were provided with either sugar or water as described above (S2 Fig). Separate cohorts of identically treated, non-blood-fed females

were included to obtain average non-blood-fed female weights for blood meal size calculations or to normalize hemoglobin concentrations for blood meal digestion experiments (see below).

## Blood meal size measurements

On the third day post-mating treatment (5–6 days PE), females held in recovery buckets (n = 50 per treatment group for two trials for a total of approximately n = 100 females per group) were offered a human forearm from the same individual (SAP) for each treatment and trial for 5 min (Cornell IRB Human Subjects Activity Exemption, FWA 00004513). Feeding was permitted for 5 min under these conditions because for females at this age range, five min was sufficient to obtain a nearly saturating (≈85–100%) feeding avidity for each treatment group (see Fig 1C). Immediately after this blood-feeding period, females were transferred to a 15 mL falcon tube and flash frozen in liquid nitrogen for 30 s to eliminate fluid loss from diuresis. Non-blood-fed females for each treatment group were also frozen to provide baseline weights for blood meal mass calculations as previously described [76]. Females were weighed individually on a Cahn C-31 microbalance to determine their mass in mg (Cahn Instruments Inc., Cerritos, CA, USA). Prior to weighing, blood-fed females were sorted on an Olympus

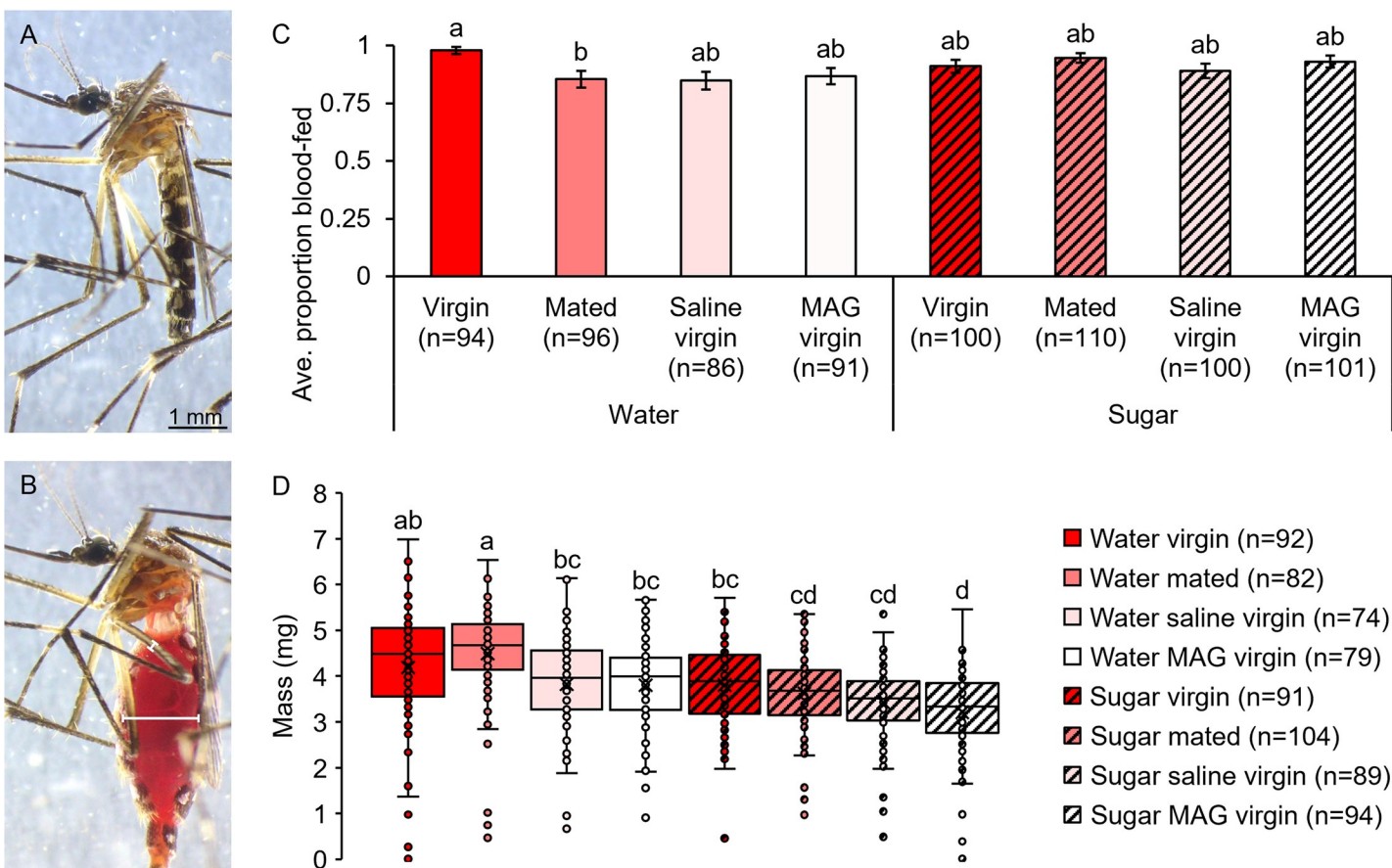

**Fig 1. Mating and MAG do not affect initial blood meal size, but sugar feeding does.** The masses of non-blood-fed (A) and blood-fed (B) females were used to determine the initial blood meal size. White bars in panel B illustrate ISD and femur width measurements used for blood meal engorgement estimates presented in S7 and S8 Figs. Feeding avidities were similar regardless of mating and sugar feeding status (C). Although mating and MAG injection did not affect blood meal size compared to non-injected and saline-injected virgin controls, prior sugar feeding resulted in smaller blood meals (D). Error bars in panel C denote SE of sample proportions and whiskers in panel D denote the minimum and maximum values. Box plots display the boundaries of the first (bottom) and third (top) quartiles, median lines, mean markers ("x"), and individual data points, including outliers. Letters above columns and box and whisker plots denote H-B-corrected post-hoc comparison p-values, with differing letters indicating significantly different groups.

SZX2-ILLT dissecting microscope (Olympus, Center Valley, PA, USA) using Sella's score (SS) criteria [80,81] into fully engorged (SS = 2) and partially engorged (SS≥3) groups, with scores in this case representing blood meal area in the abdomen rather than degree of digestion. Of females that blood fed, the average proportion that fed to full engorgement was 0.98 ± 0.01 (± standard error, or SE). Because the proportion of blood-fed females that fed to partial engorgement was only 0.02 ± 0.04 (± SE) and was not affected by combined mating treatment and sugar feeding treatment (GZLM: treatment, p = 0.644), we combined these two feeding categories into a single "blood-fed" category both here and throughout, as similar trends were observed in the other experiments (S1 Table).

### Blood meal digestion measurements

Females were blood fed as in the blood meal size experiments. To ensure that females began with equivalent amounts of blood prior to the digestion time course, partially engorged females, as assessed by visual observation, were removed immediately post-feeding. Mosquitoes were measured at eight time points post-feeding to encompass an entire gonotrophic cycle from initial blood meal ingestion to egg production and laying: 0, 8, 24, 32, 48, 56, 72 and 80 h post-blood meal (see Fig 2). For each of these eight time points, four females per

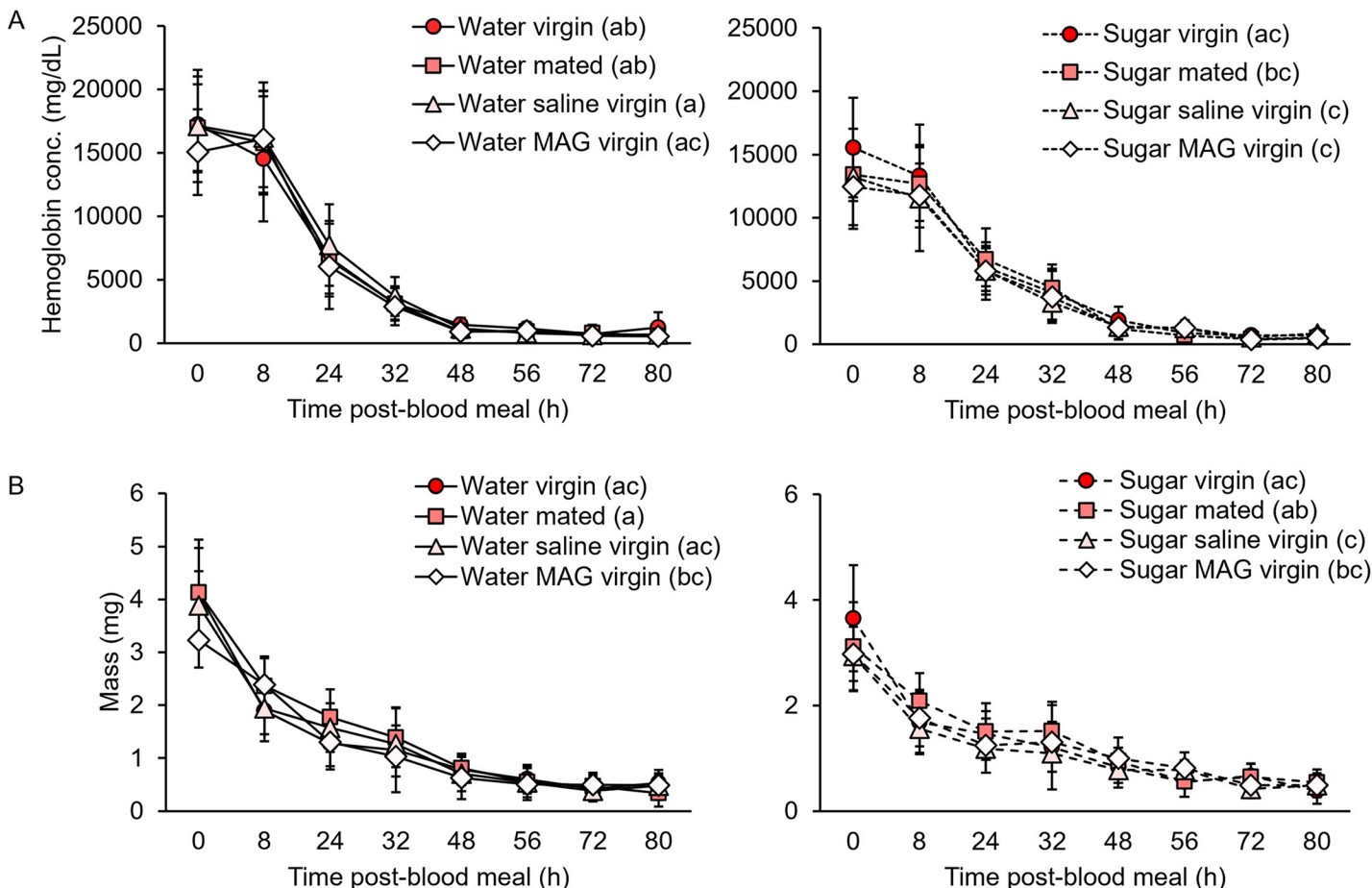

**Fig 2. Mating and MAG injection do not alter blood meal digestion in an initial feeding, but sugar feeding slows hemoglobin digestion.** (A) Neither mating nor MAG injection affected females' blood digestion rates as measured by change in total hemoglobin levels over time (n = 96 total females per group). Sugar feeding led to lower starting amounts of hemoglobin and slower digestion rates compared to water-fed groups. (B) Neither mating nor MAG injection affected female blood digestion rates as measured by change in weight over time (n = 96 total females per group). Sugar feeding led to smaller blood meals but did not affect digestion rates over time as determined by this method. Error bars denote SD. Letters in parentheses next to the treatment group names denote H-B-corrected post-hoc comparison p-values.

treatment group (n = 32 per treatment for three trials for a total of n = 96 females per group) were flash frozen using the method described above. To enable comparison of female hemoglobin content, weight, and abdominal distension over time in both laboratory and field-based specimens (see below), females were weighed, scored for degree of engorgement, and imaged under the dissecting microscope using a Sony IMX214 Exmor R CMOS 13 megapixel camera (Sony Corp., Minato City, Tokyo, Japan; See Fig 1A and 1B for examples) prior to storage at -80˚C for later processing of hemoglobin content. For imaging, a mm-scale ocular micrometer was used for calibration and later measurement of abdominal distension using a method comparable to that used for field mosquitoes (see below).

Hemoglobin was detected using a hemoglobin assay kit protocol and reagents (Sigma-Aldrich, St. Louis, MO, USA) that uses a Triton/NaOH-based method to convert hemoglobin into a colorimetric product that can be measured spectrophotometrically. Blood-fed and non-blood-fed females were homogenized for 30 s in 600 µl of water, representing a 1:150 dilution factor for fully engorged females that was found in pilot experiments to be optimal for all time points given the assay's linear detection range. Samples were then centrifuged at 10,000 g for 15 min at 4˚C and resulting supernatants were transferred to fresh microcentrifuge tubes and stored at -80˚C. Hemoglobin concentrations were calculated based on sample absorbances, which were measured using a BioTek 800 TS microplate reader (BioTek Instruments, Inc., Winooski, VT, USA).

To compare abdominal distension of females across the digestion time course with measurements of engorgement taken in the field (see below), digital measurements from images taken during this experiment were made using ImageJ software v1.52r (NIH, Bethesda, MD, USA) [82]. Images were calibrated using an ocular micrometer. Each female's inter-sclerite distance (ISD), defined as the widest point of separation between the ventral sternites and the dorsal tergites, and midleg femur width was measured (see Fig 1B). ISDs were measured digitally and divided by each female's corresponding femur width to obtain ISDs in femur widths rounded to the nearest half femur. For individuals that had no visible integument between the sternites and tergites, an ISD value of 0 was recorded.

### Blood-feeding avidity measurements

Blood-feeding avidity was defined here as the proportion of females that blood fed on a host during a given time interval. We measured blood-feeding avidity in both an initial blood meal and across multiple feedings. It is well established that most females are ready to ingest their blood meal between 12 and 24 hours after eclosion, although this timing is dependent on the ambient temperature mosquitoes experience (reviewed in [67]; [68]). Our initial feeding avidity data (see Fig 1C) derived from females on the third day post-mating treatment (5–6 days PE). However, given that *Ae. aegypti* females blood feed soon after eclosion in nature [64], but rarely in laboratory culture, we designed our feeding avidity experiments to start on the first day post-mating treatment and sugar feeding treatment (2–3 days PE; S2 Fig).

**Initial blood meal: Human host forearm.** We measured feeding avidity in an initial blood-feeding opportunity first by presenting a host forearm (SAP) for five min to mosquitoes (n = 50 per treatment for four trials for a total of approximately n = 200 females per group). Mosquitoes were held in a 6.5 L glass Pyrex beaker covered with a mesh lid and lined with two 14 cm x 14 cm black cloth resting pads (see Fig 3A and 3B). Although resting pads offered competing visual and tactile attractants, and thus contributed to lower feeding avidity (≈20–60%) compared to similar experiments (see Figs 1, 4 and 5), they were effective in standardizing mosquito resting position prior to host presentation in the larger, transparent holding containers used in these experiments. After an initial 10 min acclimation period, feedings were recorded using a FITFORT 4K Action Camera (Dreamlink E-Commerce Co., Shenzhen, GD,

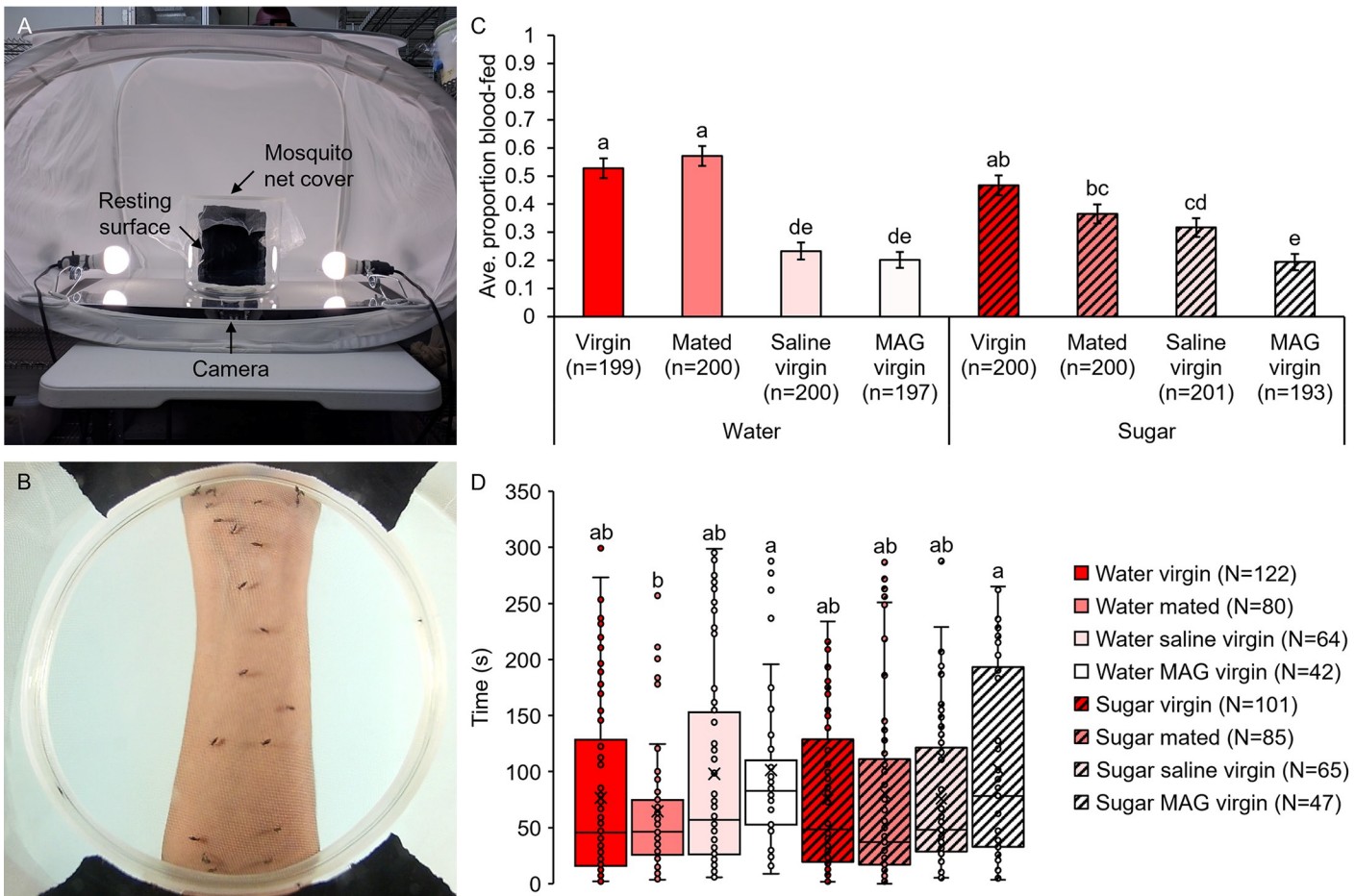

**Fig 3. Mating and MAG do not affect blood-feeding avidity or latency to feeding on a human host arm, but sugar feeding lowers avidity.** Blood-feeding avidity experiment setup (A) with an accompanying still image from a feeding latency video showing mosquitoes feeding on a host forearm (B). (C) Blood-feeding avidity was similar between non-injected virgin and mated females as well as between saline- and MAG-injected females. Injection and sugar feeding both lowered blood-feeding avidity. (D) Latency to blood feeding did not differ between treatment groups. Error bars in panel C denote SE of sample proportions and whiskers in panel D denote the minimum and maximum values. Box plots display the boundaries of the first (bottom) and third (top) quartiles, median lines, mean markers ("x"), and individual data points, including outliers. Letters above columns and box and whisker plots denote H-B-corrected post-hoc comparison p-values.

CN) placed under a 61 cm x 48 cm plexiglass stage inside a 90 cm x 64 cm x 64 cm photo studio shooting tent. Lighting for video recordings was provided by two 5.5-watt LED lights that generate minimal heat. Video analyses were performed using Adobe Premiere Pro 2020 (Adobe Inc., San Jose, CA, USA) to calculate the latency to blood feeding, defined as the time that elapsed from host presentation to the initiation of feeding for each individual female. Feeding initiation was defined as the moment when a female first landed on the host forearm, after which probing and ingestion of blood immediately commenced. Since the forearm was stationary for the duration of the assay, females that blood fed nearly always did so at same location on the host on which they initially landed. In the rare event when a female moved to reinitiate feeding at a nearby location on the host, the partially engorged female was fully trackable and the subsequent phase of feeding was excluded from latency measures.

**Initial blood meal: Whole human host.** In a separate set of experiments, we again measured feeding avidity in an initial blood meal by presenting the same entire human host (LCH) for each treatment and trial inside of a 2 m x 1 m x 1.5 m bednet for two min to a mixed group

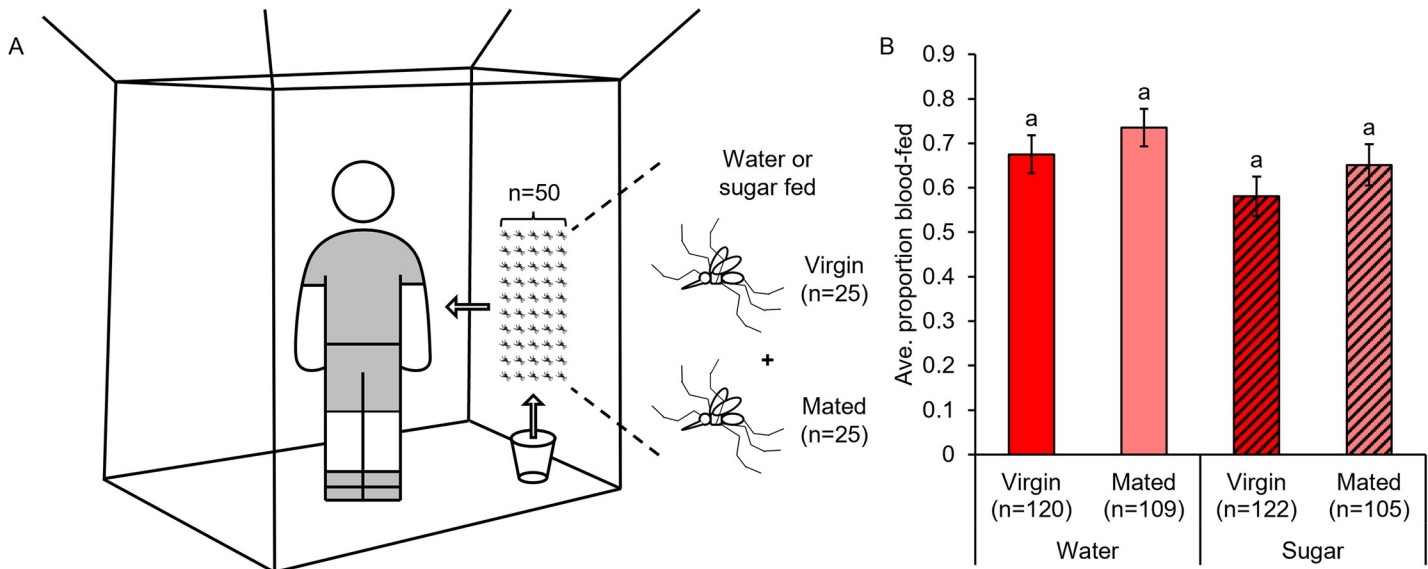

**Fig 4. Mating and MAG do not affect feeding avidity on a whole human host.** (A) Whole host feeding bednet trials experiment setup. (B) Mating treatment and sugar feeding did not affect female blood-feeding avidity. Error bars denote SE of sample proportions. Letters above columns denote H-B-corrected post-hoc comparison p-values.

(n = 50 per group for five trials for a total of approximately n = 250 females per group) of either water- or sugar-fed virgin (n = 25) or mated (n = 25) females (see Fig 4A; Cornell IRB Human Subjects Activity Exemption, FWA 00004513). Feeding was permitted for two min, rather

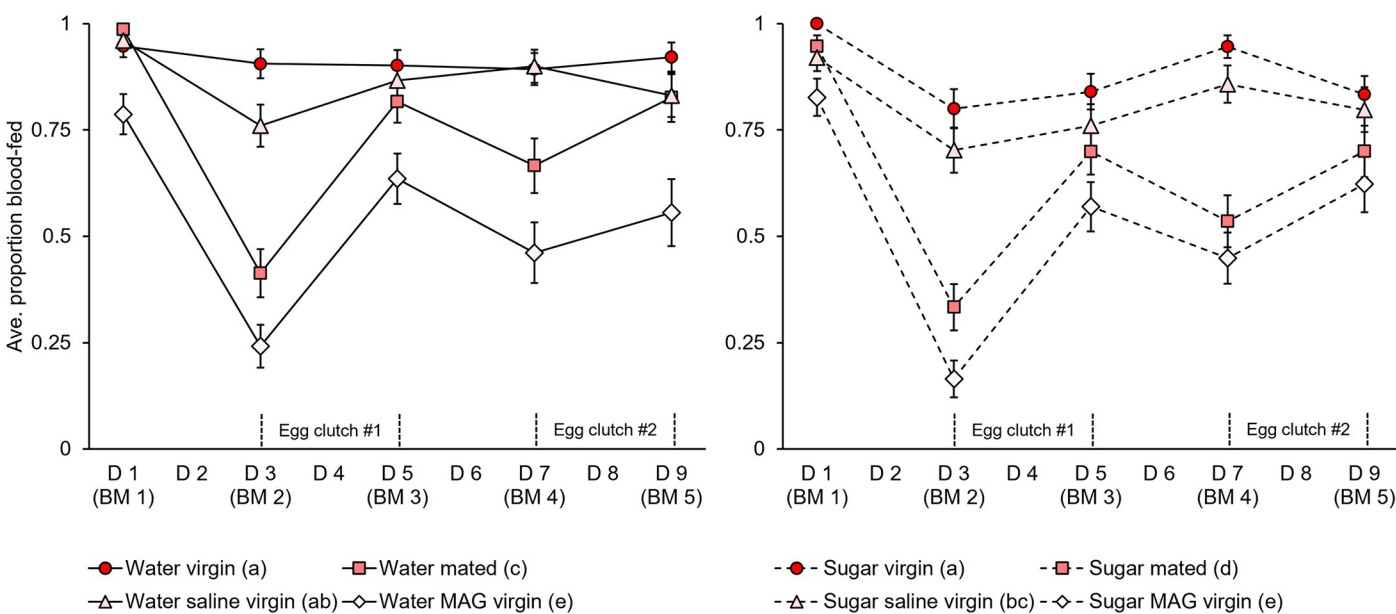

**Fig 5. Mating, MAG, and sugar feeding lower blood-feeding avidity over multiple successive meals.** Blood-feeding avidity for mated and MAG-treated females was lower compared to non-injected and saline-injected virgins, particularly in blood meals two and four (n = 75 total females per group). Egg laying activity occurred primarily between blood meals two and three (first clutch) and four and five (second clutch). Sugar feeding led to lower blood-feeding avidity over multiple blood meals compared to groups that fed on water alone. Error bars denote SE of sample proportions. For detailed egg laying data collected from trial three of these experiments, see S5 Fig. Letters in parentheses next to the treatment group names denote H-B-corrected post-hoc comparison p-values.

than for five min, to obtain potentially discriminating, but non-saturating feeding avidities (i.e., ≈60–70%) in these experiments, which featured greater host skin surface area for feeding compared to the other feeding avidity experiments. Females were released from a 0.5 L cardboard cup placed inside the bednet for an initial 5 min acclimation period prior to host presentation. At the end of the feeding interval, females were vacuum aspirated and then frozen at -20˚C to determine blood-feeding avidity and mating status (presence or absence of sperm in dissected spermathecae).

**Multiple blood meals: Human host forearm.** In a final set of experiments, we measured feeding avidity across multiple successive feedings. A portion of a host forearm (SAP) was presented for five min to mosquitoes (n = 25 per group for three trials for a total of approximately n = 75 females per group) held individually in 0.5 L cardboard cups over five sequential host feeding opportunities occurring every other day for nine days. Feeding was permitted for five min in these experiments because this provided ample opportunity for most individually-held mosquitoes of this age to obtain an initial feeding (≈80–100%) as well as a subsequent feeding. Females were fed every 48 h, as this is the time interval at which most females feed in nature during a typical gonotrophic cycle [83] as well as the time required for complete digestion of a blood meal (see Results). For all trials, females were provided with constant access to a water-soaked cotton pad on a mesh cup lid for hydration and egg laying. The first day of egg laying was noted and daily presence or absence of eggs laid for each individual was recorded in trial three.

## Field collection of mosquitoes and determination of blood meal and mating status

To determine if mating influenced blood meal size and feeding avidity in natural populations, we examined *Ae. aegypti* mosquitoes collected from domestic indoor resting sites in urban communities of Popular and Manrique in Medellín, Colombia (permission obtained from the Secretaria de Salud de Medellín). These locations were chosen based on their high dengue endemicities and *Ae. aegypti* densities. All collections were conducted with resident permission and occurred between the hours of 09:00 and 16:00. Mosquitoes were collected using Prokopack aspirators [84] operated over all accessible household surfaces. Collected mosquitoes were transferred into microcentrifuge tubes and placed on ice until further examination.

To determine blood meal size and feeding avidity, defined for field-collected mosquitoes as the proportion of females that contained blood in their abdomens at the time of collection, females with blood in their abdomens were scored for degree of engorgement as described above. Females were then binned into the following groups based upon the proportion of their abdomen that was filled with blood: 1–0.75 (SS = 2), 0.74–0.5 (SS = 3), 0.49–0.25 (SS = 4), and 0.24–0.01 (SS = 5,6) [80,81]. Females that contained fresh red blood were also recorded. ISDs of females with blood were measured to the nearest half femur using midleg femur widths analogous to the method described above as a means of quickly scaling engorgement to body size in the field. A pilot experiment conducted later in the laboratory demonstrated that hand-measured ISDs using this field-based method closely approximate laboratory-based digital measurements (S2 Table). Finally, females' spermathecae were dissected to assess mating status and ovaries were dissected to determine stage of egg development (pre-vitellogenic or vitellogenic) [85].

## Vectorial capacity calculations

We used human-biting rate and survival data derived from our laboratory and field studies to estimate the impact of combined mating treatment and sugar feeding treatment ("Thai

laboratory") or mating status ("Colombian field") on vectorial capacity, or the rate at which a mosquito transmits an infection from a currently infectious case [86]. For vectorial capacity (*C*) we used the following standard equation:

$$C = \frac{ma^2 p^n V}{-\ln p}$$

For the sake of comparison, we used uniform, biologically relevant values for all parameters except human-biting rate (the factor manipulated in our study) for both Thai and Colombian mosquitoes. For mosquito density (*m*), or the density of vectors in relation to density of hosts, we used a standard value of three vectors per host. Human-biting rates (*a*), or the number of human blood meals per vector per day, were derived from our multiple successive blood-feeding avidity trials for laboratory specimens and blood presence or absence data from field-collected specimens. Biting rates for laboratory specimens were calculated as the average daily feeding avidity of females from each treatment group multiplied by 5/9, or the biting rate if the mosquito blood fed at each of the five feedings opportunities. We also compared these laboratory-experiment-derived biting rates with a natural Thai mosquito biting rate derived from field collections ("Thai field") [78]. For mosquitoes collected in Colombia, this same procedure was used, only with feeding avidity calculated as the proportion of virgin or mated females that contained blood at the time of collection. Vector competence (*V*), or probability of acquiring an infection from an infectious person, depends on intrinsic factors, such as mosquito and virus genetics [87], as well as environmental factors, such as temperature [88], and can vary widely in *Ae. aegypti*, even amongst Thai strains [89]. We therefore selected a value of 0.8, which has been documented at 26˚C and 14 days post-exposure to infectious dengue virus blood meals in mosquito strains derived from the same field sites in Thailand as our field-derived laboratory strain [90]. For the daily probability of mosquito survival (*p*) we used daily survival values averaged across all groups that were derived from our daily mortality data (S3 Fig). To show how differences in survival due to mating, MAG injection, and sugar combined with different biting rates impact vectorial capacity (value in parenthesis in "*C*" column), we also included each treatment group's individual daily survival value (value in parenthesis in "*p*" column). For comparison to these laboratory-derived survival rates, we also included a natural Thai mosquito survival rate derived from the field (value in parenthesis in "*p*" column of "Thai field" group) [91]. For the extrinsic incubation period (*n*), or the time from ingestion of a virus to infectivity, we used 10 days, which is the average for dengue infections in *Ae. aegypti* at 26˚C [92].

## Statistical analyses

All data were analyzed using IBM SPSS Statistics software (SPSS version 24, IBM Corp., Armonk, NY). Statistical power analyses were conducted prior to experiments using data derived from pilot studies to determine ideal sample sizes for subsequent analyses [93]. For all data, normality assumptions were verified prior to analysis by comparing Pearson residuals to expected normal curves. Data are listed as averages ± standard deviation (SD) or SE throughout. Raw numerical data for all figures are included in S1 Data.

For blood meal size and feeding avidity experiments, degree of blood meal engorgement (presence or absence of a partial blood meal) and blood-feeding status (blood-fed or non-blood-fed) were analyzed using a binary logistic generalized linear model (GZLM). Female size (wing lengths), blood meal mass, hemoglobin concentration, daily mortality, feeding latency, egg laying, and ISD data were analyzed in the same manner, only using linear GZLMs. As blood feeding behavior can vary even in highly controlled settings, we accounted for effects

due to variation between experimental trials by including a trial main effect in all our models. Post-hoc comparisons of combined mating treatment and sugar feeding treatment groups (referred to as "treatment" throughout) were conducted using Holm-Bonferroni (H-B)-corrected p-values to correct for multiple comparisons. Significant differences between groups were denoted using APA-style letter subscripts. Blood-feeding and egg-development data from field samples were analyzed by mating status using a Pearson chi-squared test with a Bonferroni post-test or an independent samples T-test.

To compare blood meal digestion rates, we tested for equality in the rate of change of hemoglobin concentrations, blood meal masses, and ISDs over time between mating and sugar groups. To do so, we performed univariate general linear model (GLM) tests on blood meal measures by either mating or sugar feeding treatment with time point as a covariate [94]. Digestion rates were compared across all time points (0–80 h) as well as each interval in between (0–8 h, 8–24 h, etc.). Slopes differed across mating or sugar feeding treatments if there was a significant treatment by time point interaction.

## Results

### Mating and male accessory gland extract injection do not affect blood meal size in an initial feeding in the lab

To test whether mating or MAG affect the amount of blood a female ingests in her first blood meal, we quantified blood meal size gravimetrically by weighing non-injected virgin and mated females as well as saline- and MAG-injected virgins immediately after blood feeding (Fig 1). Although combined mating treatment and sugar feeding treatment had no impact on female blood-feeding avidity (GZLM: treatment, p = 0.060; trial, p = 0.026; Fig 1C), blood meal size was significantly affected (GZLM: treatment, p<0.0001; trial, p = 0.019; Fig 1D). However, post-hoc comparisons showed that this effect was not due to an effect of mating or MAG but rather to sugar feeding and injection treatment (see letters denoting H-B-corrected p-values; Fig 1D). Specifically, non-injected virgins and saline-injected virgins imbibed similar amounts of blood compared to mated females and MAG-injected virgins, respectively. Further, groups that fed on water or that did not receive an injection prior to blood feeding tended to ingest more blood compared to groups that fed on sugar or that received an injection. These results show that mating and MAG do not affect the amount of blood ingested in an initial feeding, while prior sugar feeding and injection treatment reduce blood meal intake.

### Mating and male accessory gland extract injection do not affect blood meal digestion rates after an initial feeding in the lab, but prior sugar feeding does

We next tested whether mating status affected blood meal digestion rates in an initial blood meal by measuring hemoglobin levels at numerous time points post-feeding (Fig 2A). Hemoglobin levels differed by treatment (GZLM: treatment, p<0.0001; trial, p = 0.063) and post-hoc comparisons showed again that this was not due to an effect of mating or MAG, but rather to sugar feeding, which depressed blood meal intake (H-B post-hoc tests). Hemoglobin levels rapidly declined over the observed gonotrophic period (GZLM: time point, p<0.0001): after an initial eight-hour period of quiescence (GLM: 0–8 h, p = 0.720), most of the blood meal was rapidly digested over the following 24 hours, with sugar-fed females digesting at lower rates compared to water-fed females (8–24 h, p = 0.006; 24–32 h, p = 0.029). By contrast, mating treatment had no effect on hemoglobin digestion rates in either water- or sugar-fed groups (GLM: 0–80 h, p = 0.735 and 0.177, respectively). This was true even during the initial phase of

blood digestion (GLM: 0–8 h, p = 0.458 and 0.836, respectively), a time frame in which an earlier study had reported a more rapid onset of digestion in mated females compared to virgins [58].

We also quantified blood meal size in these same mosquitoes by weighing to compare changes in mass and hemoglobin levels over time (Fig 2B). As with hemoglobin levels, blood meal masses declined precipitously over time (GZLM: time point, p<0.0001) and were significantly affected by treatment (treatment, p<0.001; trial, p = 0.044). Similarly, this effect was not due to mating or MAG injection, but rather again to prior sugar feeding, which caused females to imbibe less blood (H-B post-hoc tests). Overall digestion rates, as approximated by change in blood meal mass over time, did not differ by mating treatment in either water- or sugar-fed groups (GLM: 0–80 h, p = 0.114 and 0.084, respectively). However, unlike hemoglobin levels, which are not impacted by fluid loss due to diuresis, blood meal masses declined immediately post-feeding and at rates that were similar between water- and sugar-fed females (GLM: p≥0.109 for all time intervals).

Previous studies showed increased survival in mated [37,40] and MAG-injected virgin females [40] relative to non-injected and saline-injected virgins as well as in blood and sugar fed females relative to starved females [95]. Consistent with this, we found that daily mortality rates were affected by combined mating treatment and sugar feeding treatment (GZLM: treatment, p<0.0001; trial, p = 0.706; S3 Fig), decreasing due to mating or treatment with MAG, sugar feeding, and blood feeding (H-B post-hoc tests). Together, these results show that while mating and MAG do not affect blood meal digestion in an initial blood meal, they do improve female survival, and sugar feeding prior to blood feeding lowers digestion rates.

### Mating and male accessory gland extract injection do not impact host feeding avidity in an initial blood meal in the lab, but prior sugar feeding does

Although mating and MAG injection did not influence blood meal engorgement or digestion after an initial blood meal, we tested for potential post-mating effects on blood-feeding avidity, defined as the proportion of females that blood fed on a host during a given time interval. Because feeding avidity on a human host forearm did not differ between mating treatment groups under previous experimental conditions (Fig 1C), we tested female feeding avidity in an initial blood meal at an earlier post-eclosion time point in an age range in which initial blood meals commonly occur in nature (Fig 3; see Methods for details). Under these conditions, combined mating treatment and sugar feeding treatment affected blood-feeding avidity (GZLM: treatment, p<0.0001; trial, p<0.0001; Fig 3C). However, as with blood meal sizes, this effect was due to sugar feeding and injection treatment, rather than mating or MAG, as water-fed and injected females displayed lower feeding avidities compared to sugar-fed and non-injected females (H-B post-hoc tests). Thus, at earlier feeding times post-eclosion, sugar feeding and injection, but neither mating nor MAG, decreased a female's feeding avidity in an initial blood meal.

Latency to feeding, defined as the time that elapsed from host forearm presentation to feeding for each individual mosquito, varied significantly by combined mating treatment and sugar feeding treatment (GZLM: treatment, p = 0.001; trial, p<0.0001; Fig 3D). However, these differences again stemmed from injection alone and not from mating, MAG injection, or even sugar feeding (H-B post-hoc tests). Thus, we conclude that neither mating nor MAG injection affected feeding avidity or latency in an initial blood meal, while sugar feeding lowered the proportion of females that fed (avidity) but not the time to feeding (latency) of those that fed.

To test feeding avidity with a more natural host presentation, groups of mosquitoes comprised of equal numbers of either water- or sugar-fed virgin and mated females were released simultaneously into a bednet containing a human host (Fig 4A). In whole host feeding trials, combined mating treatment and sugar feeding treatment did not influence feeding avidity in an initial blood meal (GZLM: treatment, p = 0.091; trial, p<0.001; H-B post-hoc tests; Fig 4B). However, analysis of treatment groups separately by trial revealed that blood-feeding avidity in sugar-fed groups, but not in water-fed groups, increased over the experimental trials (GZLM: treatment with trial, p = 0.004; H-B post-hoc tests; S4 Fig). Taken together, these results show that in an initial blood meal, mating and MAG do not impact female feeding avidity. However due to apparent differences in sugar feeding that require further investigation, consumption of sugar, to the extent that it occurs, may impact a female's tendency to blood feed.

## Mating, male accessory gland extract injection, and sugar feeding impact host feeding avidity across multiple successive blood meals in the lab

As multiple, frequent blood meals are a well-documented feature of *Ae. aegypti* feeding habits in nature [77,78], we quantified feeding avidity on a human forearm over the course of five consecutive blood meals offered every 48 hours (Fig 5). While blood-feeding avidity was again similar across mating treatments in an initial blood meal, feeding avidity was markedly lower in mated and MAG-injected females compared to non-injected and saline-injected groups in subsequent feedings (GZLM: treatment, p<0.0001; trial, p = 0.003). Interestingly, however, mating and MAG effects on avidity were not uniform across feedings (blood meal number, p<0.0001) but rather were most pronounced in the second and fourth blood meals. The timing of these meals corresponded with the onset of peak egg laying activity for the first and second egg clutches, which occurred primarily between the second and third, and the fourth and fifth blood meals, respectively (Figs 5 and S5). Surprisingly, initial sugar feeding prior to the first blood meal led to an inhibitory effect on blood-feeding avidity that persisted even through multiple feedings (H-B post-hoc tests). Together, these feeding avidity data show that while mating and MAG do not affect feeding in an initial blood meal, the processes they trigger in females, particularly those related to egg production, correlate with a partial suppression of feeding in subsequent blood meals. Furthermore, consumption of sugar prior to blood feeding depresses blood-feeding avidity in initial as well as subsequent blood meals.

## Mating status does not affect the likelihood of containing a blood meal in a natural disease endemic field setting

To compare our blood-feeding avidity and blood meal size data collected in the laboratory to a more realistic setting, we collected mosquitoes from domestic indoor resting habitats in Medellín, Colombia. We found that an equal proportion of virgin (0.79) and mated (0.79) field-caught mosquitoes contained blood in their abdomens at the time of collection (Pearson chi-squared test: p = 0.875; Fig 6A). We also found that a significantly greater proportion of mated females (0.96) compared to virgin females (0.83) contained either blood or vitellogenic-stage eggs at time of collection (Pearson chi-squared test: p<0.0001; Fig 6B). Because females that are mated or contain vitellogenic stage eggs would on average be older than females that are virgin or do not contain mature eggs, the females we collected may have differed by age and/or gonotrophic cycle stage. These observations show that, in natural urban field settings, virgin and mated females are equally likely to contain blood, but mated females tended to be in more advanced stages of the gonotrophic cycle compared to virgins (for further details, see S6 Fig). Although this may also suggest that mated females (particularly those with substantial amounts of undigested blood and fully developed eggs) are more likely than virgins to have

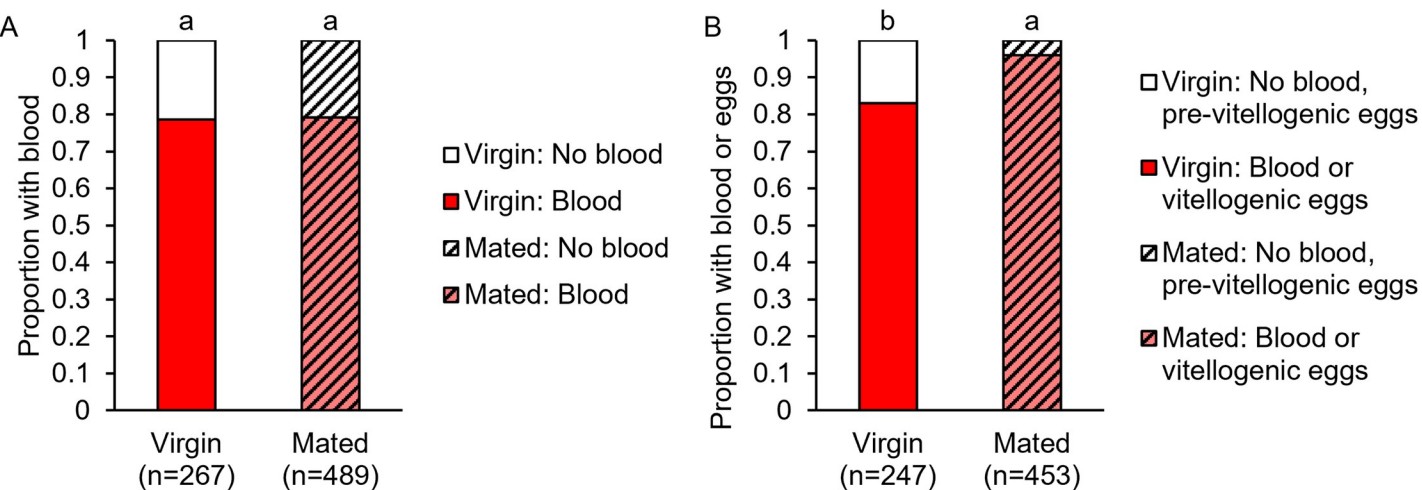

**Fig 6. Virgin and mated females collected from natural urban field settings contained blood at similar proportions but differed in reproductive life history.** (A) Virgin and mated females collected resting inside houses in Medellín, Colombia were equally likely to contain blood in their abdomens. (B) Mated females were more likely than virgins to contain blood or have vitellogenic stage eggs. Letters above columns denote Bonferroni-corrected post-hoc comparison p-values.

ingested multiple blood meals, we cannot rule out the possibility that females that contained small amounts of blood developed eggs from their current blood meal.

Finally, although measurements of blood meal engorgement in field-collected Colombian mosquitoes suggest that mated females display greater abdominal distension compared to virgins, this was not the case in Thai mosquitoes examined in the laboratory (for details, see S1 Text and S7 and S8 Figs). Together, our results indicate that under controlled conditions where egg development and other factors are accounted for, mating and MAG do not necessarily lead to increased blood meal engorgement in an initial blood meal. However, additional work is needed to understand their effects in the field.

### Mating, MAG extract, and sugar feeding could impact *Aedes* vectorial capacity estimates

We used feeding avidity data from both our Thai laboratory and Colombian field strains to calculate human-biting rates for each of our mating and sugar feeding treatment groups to determine the effect these factors might ultimately have on vectorial capacity (Table 2). These estimates show that mating, MAG injection, and sugar feeding reduce female biting rates, increase female survival rates, and hence may impact vectorial capacity estimates.

### Discussion

The transfer of male-derived seminal fluids during mating induces a series of epidemiologically important physiological and behavioral phenotypes in female mosquitoes, including resistance to courtship [27] and re-mating [8–15], increased egg development and egg laying [16–26], and increased survival [40]. Since mating has also been reported to decrease flight activity [28–33] and reduce host seeking behavior [19,34–39], and mating often occurs near hosts in *Ae. aegypti*, numerous studies have explored a potential role for mating in modulating host blood feeding. However, these prior studies have come to contradictory conclusions. To resolve these discrepancies, we utilized a series of physiological and behavioral blood-feeding assays that controlled for such previously unaccounted factors as sugar feeding, diuresis, and ambient temperature and humidity, among others, in a field-relevant *Ae. aegypti* (Thai) strain.

**Table 2. Vectorial capacity estimates in both experimental laboratory Thai strain and field-derived Colombian strain(s).**

| | | | | | | | |
|---|---|---|---|---|---|---|---|
| | | | | Vectorial capacity | | | |
| Strain | Treatment | *m* | *a* | *V* | *p* | *n* | *C* |
| Thai laboratory | Water virgin | 3 | 0.51 | 0.80 | 0.90 (0.84) | 10 | 2.05 (0.60) |
| | Water mated | 3 | 0.41 | 0.80 | 0.90 (0.89) | 10 | 1.35 (1.09) |
| | Water saline virgin | 3 | 0.48 | 0.80 | 0.90 (0.82) | 10 | 1.83 (0.41) |
| | Water MAG virgin | 3 | 0.30 | 0.80 | 0.90 (0.88) | 10 | 0.70 (0.43) |
| | Sugar virgin | 3 | 0.49 | 0.80 | 0.90 (0.97) | 10 | 1.91 (12.56) |
| | Sugar mated | 3 | 0.36 | 0.80 | 0.90 (1.00) | 10 | 1.01 (116.95) |
| | Sugar saline virgin | 3 | 0.45 | 0.80 | 0.90 (0.94) | 10 | 1.60 (3.86) |
| | Sugar MAG virgin | 3 | 0.29 | 0.80 | 0.90 (0.99) | 10 | 0.68 (11.87) |
| Thai field | | 3 | 0.76 | 0.80 | 0.90 (0.80) | 10 | 4.59 (0.67) |
| Colombian field | Blood virgin | 3 | 0.44 | 0.80 | 0.90 | 10 | 1.52 |
| | Blood mated | 3 | 0.44 | 0.80 | 0.90 | 10 | 1.54 |

Formula variable abbreviations: $m$ = density of vectors in relation to density of host; $a$ = human-biting rate, or number of human blood meals per vector per day; $V$ = vector competence, or probability of acquiring an infection from an infectious person; $p$ = daily probability of mosquito survival; $n$ = extrinsic incubation period in days; $C$ = vectorial capacity = $(ma^2p^nV)/(-\ln(p))$. Values in parentheses represent each treatment group's individual daily survival value ("$p$" column) with corresponding vectorial capacity estimates ("$C$" column) to compare these estimates with those based on averaged daily survival values (non-parenthetical values in "$p$" and "$C$" columns). Natural Thai mosquito biting and survival rates derived from the field (value in "$a$" column and value in parentheses in "$p$" column, respectively, of the "Thai field" group) were included for comparison to laboratory-derived survival values.

Further, we compared these laboratory-based observations with natural blood-feeding patterns in urban *Ae. aegypti* derived from domestic indoor resting sites in Colombia.

We observed that neither mating nor MAG injections affect blood meal intake, digestion, or feeding avidity for the first blood meal taken by female *Ae. aegypti*. Instead, our data show that sugar feeding consistently affects each of these aspects of blood feeding, which may explain the conflicting results of previous studies. We found that mating and MAG profoundly affect blood-feeding avidity differentially across multiple successive blood-feeding events. We also present field data that support these laboratory findings and suggest that feeding on human hosts in nature is largely opportunistic, as virgin and mated females were equally likely to have taken a recent blood meal at the time of collection. Overall, our results suggest that although *Ae. aegypti* feed similarly in an initial blood meal regardless of mating status, mating does have important effects on mosquito blood-feeding avidity across multiple feedings, which could ultimately impact vectorial capacity and disease transmission. Finally, sugar feeding, which is rare for *Ae. aegypti* in nature [69–75] but is a common feature of their culture in the laboratory, likely underlies at least some of the conflicting outcomes of prior studies of post-mating blood-feeding behavior.

We found that feeding avidity was similar in mated females and MAG-injected virgins compared to non-injected and saline-injected virgins in an initial blood meal but differed substantially over the course of multiple successive blood feedings (Figs 3–5). Seminal fluid proteins allow females to develop, fertilize, and lay eggs [16–26], and our data shows that mated and MAG-injected females experience similar declines in feeding avidity during periods of equivalent egg laying activity (Figs 5 and S5). Previous work from our lab on the same Thai strain studied here showed that egg production and laying is similarly elevated in mated females and MAG-injected females compared to non-injected virgins and saline-injected virgins [40]. Although work by Judson in *Ae. aegypti* showed that blood feeding stimulates equivalent egg production (though not egg laying) in virgin and mated or MAG-treated females [19], this difference likely stems from that study's use of the highly lab-adapted Liverpool strain

compared to our studies of a field-relevant Thai strain which is regularly supplemented with wild-caught mosquitoes. Taken together, we conclude that oviposition is a major determinant of *Ae. aegypti* host feeding patterns across multiple blood meals. The feeding avidity patterns that we observed may also reflect the interrelated timing and biological context of mating and blood feeding for *Ae. aegypti* in nature, with swarming males mating with females as they approach hosts to feed [39,45,48,49]. Multiple successive blood feedings in a single gono-trophic cycle are a well-documented feature of *Ae. aegypti* feeding habits in nature [77,78] and can vary by region, climate (i.e., temperature), and transmission season [78]. Our results support earlier laboratory studies in *Ae. aegypti* that show that virgins are just as likely to blood feed as mated females in an initial blood meal [37,53,96] and that document mating-induced declines in feeding avidity across multiple blood meals that correlate with egg development and egg laying (S5 Fig) [19,37,38]. It has been observed that both oogenesis and oviposition behavior inhibit mosquito blood feeding [97–99]. We also find that rearing conditions (e.g., sugar feeding; see below) impact the outcome of blood-feeding measurements. Moreover, while we were careful to control temperature and humidity in this study (all of our experiments were conducted in controlled 27°C and 85% RH conditions), most previous studies mention only rearing temperatures and humidity without specifying that feedings were also conducted in such conditions. This, too, may explain the highly variable and lower than expected feeding avidities reported in some earlier experiments [40]. Future studies in *Ae. aegypti* should examine host-associated swarm mating under the natural rearing and sugar feeding regimens employed in the present study to test for additional blood-feeding pheno-types that were not examined here. For example, although the mating habits of *Ae. aegypti* in nature, coupled with our findings in field-collected females, suggest that these effects may not ultimately impact blood-feeding avidity (see below), future work could explore potential short-term mating and courtship effects, as avidity and transcriptomics data suggests that these may temporarily dampen host-seeking [38,100].

We found that sugar feeding early in life can depress blood meal intake, digestion, and feeding avidity (Figs 1–5). Thus, sugar feeding represents an underappreciated variable that may underlie the inconsistent results of previous studies on the effect of mating on blood feeding. Additional field studies are needed to ascertain the extent to which natural sugar feeding (i.e., from fruits, nectars, honeydew, and other carbohydrate sources in the field) as opposed to arti-ficial sugar feeding (i.e., sucrose in water in the laboratory) impacts blood feeding. However, mounting studies point to an important role of mosquito carbohydrate feeding in modulating host seeking and blood-feeding behaviors in a number of mosquito species [101,102]. In nature, *Ae. aegypti* feeds almost daily on blood [78] and does not require sugar as a dietary sup-plement for energy [69]. However, access to hosts and blood meals in the laboratory is typically infrequent. Consistent with our findings, sugar-starved females under such artificial laboratory constraints display increased blood meal intake and feeding frequency when blood is offered to compensate for nutritional deficits or dehydration [57,101,103]. Although blood, rather than sugar, is required for egg production in anautogenous species, these observations make sense considering that both sugar and blood feeding can meet other energetic demands [104,105]. Importantly, although we included both water- and sugar-fed groups in all our experiments, we could not control for or quantify differences in the degree of individual female sugar meal intake during initial post-eclosion ad libitum feeding periods. The reason for this apparent variability in sugar feeding across some of our experiments is unclear (S4 Fig), partic-ularly given our constant mosquito rearing and maintenance conditions. Whatever the precise causes, these observations merit further study, including by developing a means of quantifying individual sugar consumption prior to blood feeding that does not involve destruction of the

mosquito in the process. Together, our findings suggest that future studies examining blood feeding should account for prior sugar feeding as an important potential source of variation.

Our study shows that mating and MAG injections do not impact the size of a female's initial blood meal in the laboratory (Fig 1). Earlier reports on this in the literature were contradictory, and we argue that this likely stemmed from a number of factors, including failures to control for diuresis when estimating blood meal size [56,57], differences in mosquito age and/or blood meal measurement methods [58], a lack of biological replication and statistical power [58], and differences in mosquito access to sugar prior to blood feeding. Further, as blood-feeding duration in *Ae. aegypti* can vary from under two minutes to over five minutes even among individuals from the same natural populations [106], some of the variability between previous studies of post-mating blood meal size may stem from differences in the feeding habits of the particular mosquito strain(s) under investigation. To assess the robustness and applicability of our blood meal engorgement findings across mosquito populations from two dengue-endemic locations, we tested blood meal engorgement levels in a field-relevant *Ae. aegypti* Thai strain in the laboratory and compared these results to a field-collected strain from urban indoor resting sites from Colombia. Although mated field-collected females that contained blood displayed greater abdominal distension compared to virgins, our parallel laboratory studies suggest that this may be due to differences in egg development or other factors that were not examined in our field specimens, such as sugar feeding, age, or strain genetic background. Future studies are needed to further explore these possibilities.

Using both highly sensitive hemoglobin detection and weighing methods (Fig 2), our data show that digestion of an initial blood meal occurs at the same rate regardless of mating status but is more rapid in non-sugar-fed females compared to those that were provided with a diet that contained sugar. Although we found no evidence to support earlier reports suggesting mating and MAG stimulate a more rapid onset and completion of blood meal digestion [54,55,58], our results do corroborate these same reports' observation that once blood digestion begins, it occurs at equal rates regardless of mating status. As our data show that sugar feeding modulates blood meal intake and digestion rates, and water/sugar controls were not included in these previous studies (Table 1), the reported differences in blood meal digestion due to mating may be partly attributable to variable adult nutrition prior to blood feeding. In keeping with earlier reports [60,63] we found similar overall trends when measuring blood meal digestion using chemical (hemoglobin) and gravimetric (weight) readouts. However, by tracking digestion over an entire gonotrophic cycle, we also observed important differences between these methods that stem from changes in mosquito mass, such as those related to diuresis (0–24 h) and egg development (48–80 h; Fig 2), that are detected by weighing, but not by hemoglobin assays.

Similar to our findings with Thai mosquitoes in the laboratory, our field-collected *Ae. aegypti* from Medellín, Colombia show that virgin and mated mosquitoes were equally likely to contain blood at the time of collection (Fig 6). We also found that mated females are more likely than virgins to have blood fed previously and at least partially undergone a gonotrophic cycle (Figs 6 and S6). To our knowledge, only two studies have reported virgin and mated *Ae. aegypti* blood feeding status in a field setting. Teesdale [107] recorded that a proportion of only about 0.05 of gravid females were virgins in a field study in Kenya, and since *Ae. aegypti* is anautogenous [7], these females must have taken blood meals. He also collected many virgin females via human landing capture, suggesting that these mosquitoes were attempting to feed. Pant and Yasuno [108] recorded slightly more females that were blood fed than were inseminated in a mark-release-recapture study in Thailand, suggesting that virgins do feed in the field. Thus, our findings provide valuable field evidence that corroborates these earlier claims. In nature, *Ae. aegypti* mate in host-associated swarms, and thus can feed before or after mating, depending on swarm density and host availability [49]. Combined with our laboratory

studies, this indicates that in nature, mosquitoes benefit from feeding opportunistically on human hosts, regardless of mating status [109]. This may explain why mating has not evolved to induce increased feeding behavior in *Ae. aegypti* as it has in other dipterans like *D. melanogaster* [44].

Human-biting rates derived from Thai mosquito feeding avidity data show that mating, MAG injection, and sugar feeding reduce female biting rates over multiple blood meals, and hence may reduce vectorial capacity estimates (Table 2). Further, these effects were more than offset by a disproportionate increase in vectorial capacity caused by the increased survival resulting from these very same factors. Thus, both mating and sugar feeding status may impact a mosquito's estimated ability to transmit pathogens. Given that our Colombian mosquito biting-rate estimates were calculated as the proportion of virgin or mated females that contained blood at the time of collection, and these proportions were identical, we found no difference in vectorial capacity estimates between virgin and mated field-caught females. This suggests that the effects we observed on feeding avidity in mated females over multiple blood meals in the laboratory may be overridden or asynchronous in field mosquitoes depending upon blood meal and oviposition opportunities, which would be more stochastic in natural settings than our experimental design allowed for. It should also be noted that our lab- and field-derived biting rates underestimated natural biting rates from the field [78], most likely due to underestimates of multiple host feedings and other potentially confounding factors, such as the frequency and timing of feeding in relation to blood digestion. Recent viral infection studies have shown that gut distension resulting from multiple blood meals can compromise the integrity of the midgut in a manner that increases the likelihood virus dissemination compared to estimates of dissemination taken after only a single meal [110]. Our vectorial capacity estimates help further our understanding of disease transmission dynamics by isolating mating and sugar feeding effects over multiple blood meals.

*Ae. aegypti* is a major arbovirus vector and thus a primary target of control efforts, including those involving the release of genetically modified mosquitoes [111]. Recent work in mosquitoes has uncovered common biological drivers behind mating, host-seeking, and blood-feeding behaviors [112]. Future research into female post-mating responses related to interconnected, epidemiologically impactful behaviors such as blood feeding will undoubtedly provide important biological insights that can be leveraged to create more effective vector and disease control strategies.

## Supporting information

**S1 Text. Supporting Results.**
(DOCX)

**S1 Fig. Female size as assessed by wing length was consistent across laboratory experiments.** Despite a statistically significant experiment effect (GZLM: p = 0.001), post-hoc analysis showed only minor variation in wing lengths in the blood meal digestion experiments compared to the other experiments. Whiskers denote the minimum and maximum values. Box plots display the boundaries of the first (bottom) and third (top) quartiles, median lines, mean markers ("x"), and individual data points, including outliers. Letters above box and whisker plots denote H-B-corrected post-hoc comparison p-values.
(TIF)

**S2 Fig. Graphical timeline of sugar and mating/injection treatments in blood feeding experiments.** Abbreviation: D, day.
(TIF)

**S3 Fig. Mating, MAG, blood feeding, and sugar feeding decrease daily mortality.** These results confirm earlier reports of increased survival due to mating [37,40], MAG injection [40], as well as blood and sugar feeding [94]. Error bars denote SE. Letters above columns denote H-B-corrected post-hoc comparison p-values.
(TIF)

**S4 Fig. Blood-feeding avidity varied across whole host feeding trials in sugar-fed, but not water-fed groups.** Analyzing combined mating treatment and sugar feeding treatment groups separately by trial showed that sugar-fed, but not water-fed female feeding avidity increased over the course of our experimental trials. Letters above columns denote H-B-corrected post-hoc comparison p-values.
(TIF)

**S5 Fig. Mated and MAG-treated females reach their peak egg laying activity between the second and third as well as the fourth and fifth blood meals.** Detailed egg laying data collected from trial three of the multiple blood meal feeding avidity experiments (n = 75 total females per group) showed that combined mating treatment and sugar feeding treatment (GZLM: treatment, $p<0.0001$) and gonotrophic cycle day (day, $p<0.0001$) significantly affected oviposition behavior. Similar trends were observed, but not quantified, for trials one and two. For comparison of egg clutch one and two egg laying peaks with feeding avidity data, see Fig 5. Letters in parentheses next to the treatment group names denote H-B-corrected post-hoc comparison p-values. Abbreviations: D, day; BM, blood meal.
(TIF)

**S6 Fig. Blood feeding and egg development of field-collected Colombian strain(s).** (A) Proportions of virgin and mated females differed significantly between combined blood and egg categories (Pearson chi-squared test: $p<0.0001$). (B) A disproportionate number of blood-fed and non-blood-fed virgins compared to mated females contained pre-vitellogenic eggs (BPV and NBPV, respectively) at the time of collection, indicating that they had likely not blood fed previously and had not yet completed a gonotrophic cycle. A disproportionate number of blood-fed and non-blood-fed mated females compared to virgins contained vitellogenic eggs (BV and NBV, respectively) at the time of collection, suggesting that many had likely blood fed previously and had at least partially undergone a prior gonotrophic cycle. Abbreviations: see far-left table column in B. Numbers above columns in B represent sample sizes. Letters above columns denote Bonferroni-corrected post-hoc comparison p-values. Comparisons in (A) are between stacked columns and comparisons in (B) are within categories.
(TIF)

**S7 Fig. Blood-fed mated females display greater abdominal distension compared to blood-fed virgins in field-collected Colombian mosquitoes, but not in Thai mosquitoes in the laboratory.** (A–C) For field-derived mosquitoes, female spermathecae were dissected to assess mating status. Mated field-collected mosquitoes from Medellín, Colombia displayed wider abdomens compared to virgins (A), even shortly after a blood meal as indicated by the presence of fresh blood (B). Virgin and mated females with pre-vitellogenic stage eggs differed in degree of abdominal distension, whereas virgin and mated females with vitellogenic stage eggs displayed similar degrees of distension (C). Laboratory experiments involving nulliparous Thai mosquitoes that had not previously blood fed (D) show that abdominal distension after an initial blood meal is not affected by mating or MAG injection (n = 96 total females per group). Error bars denote SD. Letters above columns in A and C and in parentheses next to the treatment group names in D denote H-B-corrected post-hoc comparison p-values and

letters above columns in B denote independent samples T-test comparison p-values.
(TIF)

**S8 Fig. In the laboratory, blood-fed virgin and mated females display similar degrees of abdominal distension.** (A, B) Laboratory experiments involving nulliparous Thai mosquitoes that had not previously blood fed show that abdominal distension is generally similar between combined mating treatment and sugar feeding treatment groups in an initial blood meal. ISDs were similar whether compared by the proportion of the abdomen that was filled with blood (A) or by abdomens with fresh blood (B). Error bars denote SD. Letters above columns denote H-B post-hoc test comparison p-values.
(TIF)

**S1 Data. Raw numerical data for all figures.**
(XLSX)

**S1 Table. Females most often blood fed to full, rather than partial engorgement.** Values are displayed as average ± SE. For mating and sugar effects, GZLM p-values for each set of experiments are displayed. Abbreviations: BM, blood meal.
(XLSX)

**S2 Table. Field and lab-based measurements of blood meal engorgement using femur widths are comparable.** All treatments have minimum n = 27. Values are displayed as average ± SD.
(XLSX)

# Acknowledgments

We acknowledge the Secretaria de Salud de Medellín, who granted permission to perform the field study, as well as Guillermo Rúa-Uribe at the Universidad de Antioquia, who helped facilitate the study. We thank Elisabeth Martin, Kevin Pritts, Henry Goldsmith, and David Braunstein for help with mosquito rearing and maintenance, Dr. Alex Amaro and Lindsay Baxter for technical assistance, and Dr. Lynn Johnson and Dr. Erika Mudrak of the Cornell Statistical Consulting Unit for advice on statistical analyses. Finally, we acknowledge the Research Experience for Peruvian Undergraduates Program for facilitating Stefano Garcia Castillo's internship at Cornell.

# Author Contributions

**Conceptualization:** Garrett P. League, Ethan C. Degner, Yassi Hafezi, Erica Tennant, Catalina Alfonso-Parra, Frank W. Avila, Mariana F. Wolfner, Laura C. Harrington.

**Data curation:** Garrett P. League, Ethan C. Degner, Sylvie A. Pitcher, Priscilla C. Cruz, Raksha S. Krishnan, Stefano S. Garcia Castillo, Catalina Alfonso-Parra.

**Formal analysis:** Garrett P. League, Ethan C. Degner, Laura C. Harrington.

**Funding acquisition:** Frank W. Avila, Mariana F. Wolfner, Laura C. Harrington.

**Investigation:** Garrett P. League, Ethan C. Degner, Sylvie A. Pitcher, Yassi Hafezi, Erica Tennant, Priscilla C. Cruz, Raksha S. Krishnan, Stefano S. Garcia Castillo, Catalina Alfonso-Parra, Laura C. Harrington.

**Methodology:** Garrett P. League, Ethan C. Degner, Sylvie A. Pitcher, Yassi Hafezi, Erica Tennant, Priscilla C. Cruz, Raksha S. Krishnan, Stefano S. Garcia Castillo, Catalina Alfonso-Parra, Frank W. Avila, Mariana F. Wolfner, Laura C. Harrington.

**Project administration:** Garrett P. League, Ethan C. Degner, Sylvie A. Pitcher, Yassi Hafezi, Erica Tennant, Priscilla C. Cruz, Raksha S. Krishnan, Stefano S. Garcia Castillo, Catalina Alfonso-Parra, Frank W. Avila, Mariana F. Wolfner, Laura C. Harrington.

**Resources:** Frank W. Avila, Mariana F. Wolfner, Laura C. Harrington.

**Supervision:** Garrett P. League, Yassi Hafezi, Catalina Alfonso-Parra, Frank W. Avila, Mariana F. Wolfner, Laura C. Harrington.

**Validation:** Garrett P. League, Ethan C. Degner, Yassi Hafezi, Erica Tennant, Mariana F. Wolfner, Laura C. Harrington.

**Visualization:** Garrett P. League, Ethan C. Degner, Yassi Hafezi, Erica Tennant, Priscilla C. Cruz, Raksha S. Krishnan, Mariana F. Wolfner, Laura C. Harrington.

**Writing – original draft:** Garrett P. League, Ethan C. Degner, Yassi Hafezi, Erica Tennant, Mariana F. Wolfner, Laura C. Harrington.

**Writing – review & editing:** Garrett P. League, Ethan C. Degner, Sylvie A. Pitcher, Yassi Hafezi, Erica Tennant, Priscilla C. Cruz, Raksha S. Krishnan, Stefano S. Garcia Castillo, Catalina Alfonso-Parra, Frank W. Avila, Mariana F. Wolfner, Laura C. Harrington.

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
