## [Decision Letter · Decision Letter 0]

20 Jun 2021

Dear Dr. Harrington,

Thank you very much for submitting your manuscript "The impact of mating and sugar feeding on blood-feeding physiology and behavior in the arbovirus vector mosquito Aedes aegypti" for consideration at PLOS Neglected Tropical Diseases. As with all papers reviewed by the journal, your manuscript was reviewed by members of the editorial board and by several independent reviewers. In light of the reviews (below this email), we would like to invite the resubmission of a significantly-revised version that takes into account the reviewers' comments. 

We cannot make any decision about publication until we have seen the revised manuscript and your response to the reviewers' comments. Your revised manuscript is also likely to be sent to reviewers for further evaluation.

Sincerely,

Elvina Viennet, PhD

Deputy Editor

Reviewer's Responses to Questions

**Key Review Criteria Required for Acceptance?**

**Methods**

-Are the objectives of the study clearly articulated with a clear testable hypothesis stated?

-Is the study design appropriate to address the stated objectives?

-Is the population clearly described and appropriate for the hypothesis being tested?

-Is the sample size sufficient to ensure adequate power to address the hypothesis being tested?

-Were correct statistical analysis used to support conclusions?

-Are there concerns about ethical or regulatory requirements being met?

Reviewer #1: Overall well presented.

Reviewer #2: Previous studies on the effects of mating and feeding on blood feeding volume avidity and digestion in Aedes aegypti mosquitoes have shown conflicting results. The goal of this study is to test the hypothesis that poorly-controlled nutritional variables underlie these discrepancies by carefully controlling mating status (and exposure to MAGs) and feeding conditions to identify their interactions and effects on blood feeding. The study design and population are appropriate for testing this hypothesis. The sample sizes are sufficient and statistical tests and controls are appropriate. There are no concerns about ethical or regulatory requirements.

Reviewer #3: The methods are acceptable. The one specific comment is were the mosquitoes acquired from a single cage and how were the 96 analyzed, such as were these measured in smaller cohorts from multiple cages. Please clarify as we get odd cage effects and mosquitoes from the same cage could more of less be considered technical rather than biological replicates.

**Results**

-Does the analysis presented match the analysis plan?

-Are the results clearly and completely presented?

-Are the figures (Tables, Images) of sufficient quality for clarity?

Reviewer #1: Figures and Tables overall clear. Analysis matches analysis plan.

Reviewer #2: The authors report that, although mating status does not affect blood feeding in the first meal, it does have an effect on female survival and can affect subsequent meals. Sugar feeding, a common laboratory practice for maintaining adult mosquitoes, does affect blood feeding and likely accounts for discrepancies between previous studies. Virgin and mated field-collected females from Colombia are equally likely to contain blood, indicating that females blood feed in the wild independent of their mating status. These findings clarify longstanding conflicting results and are relevant for researchers studying many components of mosquito behavior and physiology, however some of the key findings are obscured by the current presentation of the data (for example water vs sugar fed differences compared directly).

Reviewer #3: The results are clear, but I would suggest moving the vectorial capacity section to the end of the results. Also, the marking of statistical significance on the figures needs clarification. As an example, Figure 4 shows no differences, but sugar feeding is discussed as different. I believe this stems from using a basic letter system, rather that bars showing p values to highlight the specific comparisons. This would improve clarity.

**Conclusions**

-Are the conclusions supported by the data presented?

-Are the limitations of analysis clearly described?

-Do the authors discuss how these data can be helpful to advance our understanding of the topic under study?

-Is public health relevance addressed?

Reviewer #1: Conclusions are largely supported by the results presented.

Reviewer #2: These findings clarify longstanding conflicting results and are relevant for researchers studying many components of mosquito behavior and physiology, however some of the key findings are obscured by the current presentation of the data (for example water vs sugar fed differences compared directly). The calculations for vectorial capacity were not well integrated into the paper. This could either be removed or tied to the findings or predicted outcomes to more directly address public health relevance. Some of the claims (for example, in Figure S3 that variability in sugar consumption leads to variability in host-seeking) could be strengthened by quantification of sugar consumption and the major findings could be more clearly communicated through modifications to the figures and text.

Reviewer #3: Conclusions are sufficient. I'd removed the references to figures and such in the discussion, as these are included in the results and having them repeated is not necessary..

**Editorial and Data Presentation Modifications?**

Reviewer #1: (No Response)

Reviewer #2: Minor points:

Lines 84 – 87 – “In Ae. aegypti, the transfer of male accessory gland fluid (i.e., seminal fluid) proteins to females during mating renders females resistant to subsequent male mating attempts [8–15], increases egg development and oviposition [16,17,26,18–25], inhibits courtship acoustic harmonization [27], decreases flight activity [28–33], reduces host seeking behavior [18,34–39], increases survival [40], and induces immune responses [41,42].

Figure 2c and d – Should this y axis read mg (instead of g)?

Figure 4a – There is a group referred to as “sugar water fed” and this is the only instance of this usage (presumably to refer to “sugar fed”).

Table 1 – Typo: Seaton and Lumsden Timing of feeding PE should be 3 – 4 d PE

Reviewer #3: Minor revisions required.

**Summary and General Comments**

Reviewer #1: In this study, the authors examined the effects of mating on blood feeding by Aedes aegypti females: a topic that has been examined by a number of authors historically but which also has yielded varying results. This authors used a combination of controlled laboratory experiments (using a Thai-derived culture) and field data from Columbia and Thailand that together looked at a range of variables that could affect blood meal size and feeding avidity. 

Main findings are that mating has no effect on a first blood meal in regard to size, digestion rates or feeding avidity but does affect survival and also reduces blood feeding avidity over successive feeding opportunities due likely mated females producing and laying mature eggs while unmated females cannot. In contrast, sugar feeding reduced first blood meal size and interacted with mating status to affect survival and feeding avidity over successive blood feeding opportunities. Field data overall supported these trends although some differences were also found in the field that could not be fully reconciled by follow-up laboratory studies.

I overall thought the study was well-written with careful attention given to fully explaining most methods and data analysis. Figures/tables were also mostly clear. That sugar feeding affects blood feeding, while not unexpected, is nonetheless a valuable contribution to the mosquito blood-feeding literature. I only have a small number of queries or suggestions for revision.

1. Page 22, Fig. 3 versus Page 25 Fig. 5. The former presents data supporting that sugar but not mating or MAG affect avidity for an initial blood meal while the latter presents data that mating and MAG do affect avidity over successive meals. What causes some pause is the seemingly large differences in the proportion of females that initially fed between the two figures with 50-60% of virgin and mated females and 20-30% of saline and MAG females feeding in Fig. 3 while more than 90% of virgin and mated females and >75% of saline and MAG females fed at BM1 in Fig. 5. I recognize blood feeding is often variable with mosquitoes no matter how hard one tries to standardize conditions. But the data presented also makes it difficult to fully buy into concluding that mating and MAG affected outcomes over successive blood feeding opportunities when Fig. 3 shows responses that are so much lower than for BM1 in Fig. 5 but are fairly similar for the MAG-water/sugar treatment for BM2. Some comment/discussion about variation is thus potentially warranted in regard to the outcomes presented for blood-feeding avidity.

2. The Discussion section of the manuscript talks about mating reducing blood-feeding avidity over successive meals being linked to egg development and egg laying, which the authors present some results for in Fig. S4. My suggestion is that the authors elaborate a bit more on what they present in S4 and write on page 31. Specifically, several studies in the literature indicate that blood feeding stimulates egg maturation (vitellogenesis) in both virgin and mated females but virgin females very much differ from mated females in regard to oviposition with the latter usually laying many eggs in clutch and the former laying few or no eggs. Judson (1967) that the authors cite in the manuscript further report that implantation of accessory glands stimulated virgins to oviposit the same number of eggs as mated females. In contrast, I do not know if the MAG prep the authors inject similarly stimulates virgins to oviposit comparable numbers of eggs as mated females because Fig. S4 doesn’t show this and I haven’t seen results elsewhere that clearly indicates MAG preps yield the same outcome as accessory gland implantation did in Judson’s assays. So, this point I think needs to be addressed in the manuscript. The other small suggestion, if MAG injection causes virgins to oviposit similar numbers of eggs, is that the data in Fig. 5 seems to suggest mating/MAG -induced declines in feeding avidity are tied more to egg laying than egg development since virgin females that blood feed produce as many mature eggs as mated females. If MAG injection does not stimulate virgins to lay a similar number of eggs as virgins, then something other than oviposition is at play to yield the blood feeding patterns shown in Fig. 5. 

3. Fig. 6 and S5. The pie charts don’t seem essential in this figure as a reader can glean the numbers of individuals in each outcome from the N values at the bottom of the bar graphs. If the pie charts were eliminated, the authors could also perhaps make Fig. 6A, B with B showing what is currently presented in S5 (thus eliminating S5). In this way, it would be easier on a reader to see how presence of blood and vitellogenic eggs in the ovaries relate to one another. Maybe I don’t fully understand what’s written on page 26 but it would seem to me difficult to know from the data presented that the mated females were actually collected at different ages/or stages of their reproductive life histories (or had more likely blood fed previously-line 732) based on what is presented in Fig. 6 and S5. If the authors are confident this is the case, rewording is needed to clarify the reasoning why the data presented support this.

4. Fig. 7. I appreciate that the authors had hoped abdomen distention could be used a proxy for assessing blood feeding amounts, but the confounding effects of egg development and other factors make the outcomes of this part of the manuscript uncertain and also raise questions about using this measure for assessing blood meal sizes in the field. So, it may be more appropriate to shorten this section and move Fig. 7 to the Supp. Files. 

5. Line 688. I am familiar with refs 103 and 104. I agree sugar and blood feeding can both meet energetic/maintenance demands but I respectfully disagree if the authors are suggesting sugar and blood feeding can meet similar demands when it comes to egg production as I’ve never seen that sugar feeding stimulates vitellogenesis or oviposition to anything remotely comparable to blood feeding in aegypti or other anautogenous species.

6. The authors may want to consider presenting their vectorial capacity estimates as part of the Results rather as a point of Discussion given the methods used for the calculations presented are presented in the Methods section and rely on data generated in earlier parts of the study. Line 755, the word across looks like it should be deleted.

Reviewer #2: These findings clarify longstanding conflicting results and are relevant for researchers studying many components of mosquito behavior and physiology, however some of the key findings are obscured by the current presentation of the data (for example water vs sugar fed differences compared directly). The calculations for vectorial capacity were not well integrated into the paper. This could either be removed or tied to the findings or predicted outcomes to more directly address public health relevance. 

Some of the claims (for example, in Figure S3 that variability in sugar consumption leads to variability in host-seeking) could be strengthened by quantification of sugar consumption and the major findings could be more clearly communicated through modifications to the figures and text.

Lines 98 – It would be helpful to mention here that blood feeding and mating can happen in either order for successful reproduction and that virgins are capable of developing eggs and oviposition (as shown in Figure S4).

Line 143 – A more detailed explanation of “water-fed controls” would be helpful here. Does this refer to females who were provided exclusively water as adults with no access to sugar?

Line 199 – 202 Why was the male:female ratio changed between these experiments? Also it’s not clear if males were removed for the 2-3 day period when mosquitoes were held prior to feeding. 

Line 208 – A graphical timeline of the protocols starting with mosquito eclosion would help the reader visualize the order and timing of injection/mating and feeding – and this would be especially helpful in comparing the blood meal size vs avidity experiments.

Line 215 – Was mating status verified for females used in the meal size and digestion experiments?

Line 268 - 270 The authors choose to test 2 - 3 days PE timepoint for feeding to mimic more naturalistic access to hosts. These feeding rates in the experiments shown in Figure 3 are lower compared to those in their other experiments – could this be due to immature females or is this related to assay design?

Line 279 - 281 – The authors state that “Video analyses were performed using Adobe Premiere

Pro 2020 (Adobe Inc., San Jose, CA, USA) to calculate the latency to blood feeding, defined as

the time that elapsed from host presentation to the initiation of feeding for each individual female.” but it’s not clear that they can distinguish individual females to be sure that each female only fed one time during the assay?

Line 281 – How was “feeding initiation” scored? Was this landing on the forearm, skin penetration, or presence of blood/abdominal distension?

Line 295 – What was the rationale for choosing 48 hours for refeeding timepoints?

Figure 2 – Since sugar feeding seems to have the largest effect, it would be more intuitive to see direct comparisons of between water vs sugar (2a vs 2b) and to show direct comparisons between the relevant metrics for digestion/clearance rates.

Lines 427 - 433 – There appear to be conflicting statements in the legend vs title of this figure: that sugar feeding slows digestion rates using hemoglobin measurements but that sugar feeding does not affect digestion rates according to mass measurements. Is it correct that the hemoglobin assay but not the weights support the title of this figure?

Figure 5 – Adding time to the x axis would help to clarify these results. It would also be helpful to directly visualize water vs sugar fed.

Figure 5 – Were females given constant access to egg laying substrate throughout this assay?

Figure 7a – It would be helpful to see some pictures of these measurements. 

Figure S3 – Quantification of sugar consumption among the sugar-fed groups would be helpful to contextualize the variation in blood-feeding avidity here. It would be helpful to give some quantification of the range of meal sizes/calories consumed.

Table S3 – The calculations for vectorial capacity are not well-integrated into the text. I would find more discussion of the predicted differences between the Thai field and Colombian strains to be useful or a prediction of outcomes for disease transmission in these sites.

Reviewer #3: (No Response)

PLOS authors have the option to publish the peer review history of their article (what does this mean?). If published, this will include your full peer review and any attached files.

Reviewer #1: No

Reviewer #2: No

Reviewer #3: No
---

## [Decision Letter · Decision Letter 1]

14 Sep 2021

Dear Dr. Harrington,

We are pleased to inform you that your manuscript 'The impact of mating and sugar feeding on blood-feeding physiology and behavior in the arbovirus vector mosquito Aedes aegypti' has been provisionally accepted for publication in PLOS Neglected Tropical Diseases.

Best regards,

Elvina Viennet, PhD

Deputy Editor

Reviewer's Responses to Questions

**Key Review Criteria Required for Acceptance?**

**Methods**

-Are the objectives of the study clearly articulated with a clear testable hypothesis stated?

-Is the study design appropriate to address the stated objectives?

-Is the population clearly described and appropriate for the hypothesis being tested?

-Is the sample size sufficient to ensure adequate power to address the hypothesis being tested?

-Were correct statistical analysis used to support conclusions?

-Are there concerns about ethical or regulatory requirements being met?

Reviewer #1: Methods are acceptable

Reviewer #2: The objectives are clearly presented and the methods are acceptable.

Reviewer #3: (No Response)

**Results**

-Does the analysis presented match the analysis plan?

-Are the results clearly and completely presented?

-Are the figures (Tables, Images) of sufficient quality for clarity?

Reviewer #1: Results are acceptable

Reviewer #2: The analysis is clear and the authors' revisions have improved the clarity of the results.

Reviewer #3: (No Response)

**Conclusions**

-Are the conclusions supported by the data presented?

-Are the limitations of analysis clearly described?

-Do the authors discuss how these data can be helpful to advance our understanding of the topic under study?

-Is public health relevance addressed?

Reviewer #1: Conclusions are overall supported by the results presented.

Reviewer #2: The conclusions are supported by the data presented and the authors have clearly addressed the public health relevance of this work.

Reviewer #3: (No Response)

**Editorial and Data Presentation Modifications?**

Reviewer #1: (No Response)

Reviewer #2: (No Response)

Reviewer #3: (No Response)

**Summary and General Comments**

Reviewer #1: The authors have addressed the points for suggested revision that I raised after reading the original submission. I am fully satisfied with the revised ms.

Reviewer #2: The authors have appropriately addressed the reviewers' concerns.

Reviewer #3: The authors have addressed my previous comments.

PLOS authors have the option to publish the peer review history of their article (what does this mean?). If published, this will include your full peer review and any attached files.

Reviewer #1: No

Reviewer #2: No

Reviewer #3: No

---

## [Editor Report · Acceptance letter]

28 Sep 2021

Dear Dr. Harrington,

We are delighted to inform you that your manuscript, "The impact of mating and sugar feeding on blood-feeding physiology and behavior in the arbovirus vector mosquito *Aedes aegypti*," has been formally accepted for publication in PLOS Neglected Tropical Diseases.

Best regards,

Shaden Kamhawi

co-Editor-in-Chief

Paul Brindley

co-Editor-in-Chief
